# Learning Intractable Multimodal Policies with Reparameterization and Diversity Regularization

**Ziqi Wang**     **Jiashun Liu**     **Ling Pan**[*]

Hong Kong University of Science and Technology

## Abstract

Traditional continuous deep reinforcement learning (RL) algorithms employ deterministic or unimodal Gaussian actors, which cannot express complex multimodal decision distributions. This limitation can hinder their performance in diversity-critical scenarios. There have been some attempts to design online multimodal RL algorithms based on diffusion or amortized actors. However, these actors are intractable, making existing methods struggle with balancing performance, decision diversity, and efficiency simultaneously. To overcome this challenge, we first reformulate existing intractable multimodal actors within a unified framework, and prove that they can be directly optimized by policy gradient via reparameterization. Then, we propose a distance-based diversity regularization that does not explicitly require decision probabilities. We identify two diversity-critical domains, namely multi-goal achieving and generative RL, to demonstrate the advantages of multimodal policies and our method, particularly in terms of few-shot robustness. In conventional MuJoCo benchmarks, our algorithm also shows competitive performance. Moreover, our experiments highlight that the amortized actor is a promising policy model class with strong multimodal expressivity and high performance. Our code is available at `https://github.com/PneuC/DrAC`

## 1   Introduction

Despite the remarkable progress in reinforcement learning (RL) for continuous control and decision-making tasks [23, 55, 43], learning multimodal policies remains challenging [20, 33, 48]. However, most state-of-the-art RL algorithms predominantly employ deterministic or unimodal Gaussian policies [29, 39, 11, 14], which limits their ability to capture the complex, multimodal decision distributions. However, real-world applications can have multiple goals, making multimodality and decision diversity essential [13, 28, 41], where unimodal policies can be brittle to perturbations in the environment. For example, given a navigation task in a maze with multiple valid paths, unimodal policies typically converge to a single shortest route, creating a critical vulnerability – when this route becomes obstructed during deployment, such policies fail catastrophically [26]. In contrast, a multimodal policy with high decision diversity can learn varied navigation strategies, allowing it to adapt when faced with unexpected obstacles. In zero-sum games, strategic diversity that requires multimodal policies is also critical for agents to maintain robustness against adaptive opponents [31]. Additionally, RL

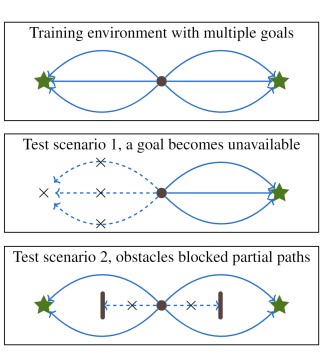

Figure 1: Motivational examples. A multimodal policy with maximized diversity can achieve multiple goals and enable robustness against environmental changes.

---

[*]Correspondence to: Ling Pan (lingpan@ust.hk)

39th Conference on Neural Information Processing Systems (NeurIPS 2025).

has been applied to generative tasks in recent years [1], where multimodal policies can potentially improve the diversity of generated objects with less harm to quality [50].

The main challenge of learning expressive multimodal policies lies in their intractability. Existing online RL algorithms that train multimodal policies are typically based on the maximum-entropy RL framework [14]. However, entropy regularization encourages policies to be more uniform, but do not specifically prefer multimodality. Meawhile, computing entropy requires an analytical expression of decision probability, which is not directly applicable to intractable actors. Some prior works attempt to mitigate this problem with a workaround that employs tractable but less expressive multimodal actors [32, 2, 50], sacrificing expressivity. Another line of research leverages Stein variational gradient descent (SVGD) as a gradient estimator [13] or as a sampler that directly samples actions from the value distribution learned by the critic [33]. However, we empirically observe that it either underperforms traditional algorithms or incurs significant computation costs. Besides, there have been some online RL algorithms training diffusion actors [53, 36, 48], but to our best knowledge, only the diffusion actor-critic with entropy regulator (DACER) [48] takes decision diversity into account among these algorithms. DACER controls diversity by scaling an additional unimodal Gaussian noise instead of optimizing diversity through gradients w.r.t. the actor's parameters.

In this paper, we propose a novel diversity-regularized RL framework to address the aforementioned challenge. First, this paper shows that existing multimodal actors can be viewed as a combination of a parameterized mapping function with a fixed latent random distribution, which we call *stochastic-mapping actors*. We then highlight that the policy gradient of any stochastic-mapping actor can be estimated via the reparameterization trick, without accessing the gradient of decision probabilities. This provides a general methodology to optimize intractable stochastic-mapping actors. Secondly, we propose a distance-based diversity regularization instead of entropy, which does not explicitly require the decision probability. Combining with the policy regularization theorems [48], we develop a novelty actor-critic algorithm named Diversity-regularized Actor Critic (DrAC). Main contributions of this paper are summarized as follows

1. We formulate a class of actors named *stochastic-mapping actors*, bridge policy gradient and intractable stochastic actors via reparameterization trick. This formulation serves as a useful tool for understanding and designing multimodal RL models and algorithms.

2. We propose a learning framework with a distance-based diversity regularization, which simultaneously optimizes expected return and decision diversity without requiring explicit access to decision probabilities.

3. We identify two domains where multimodality and diversity are critical, and conduct comprehensive experiments, including performance in standard MuJoCo benchmarks. In addition to justifying the advantages of multimodal policies and our algorithm, we also present new insights on multimodality and diversity.

## 2 Related Works

**Policy Diversity in RL**   Decision diversity (or randomness) is an essential ingredient of online deep RL algorithms [29, 11, 14], which not only enforces exploration but also reduces variance [11]. Some research has shown that decision diversity can improve sample efficiency or final performance regarding expected return [16, 34, 19, 54, 6]. Furthermore, diversity can serve in pretraining. Eysenbach et al. proposed that training policies with only a diversity objective can discover useful skills and serve as an effective pretraining method [9]. Ying et al. pretrain diffusion actors with intrinsic reward to discover diverse behaviors and fine-tune diffusion actors to fast generalize to downstream tasks [56]. Besides, Zhang et al. propose an algorithm to find varied policies for the same task [57]. Kumar et al. train multiple policies with diverse behaviors, and highlight that diversity is the key to ensuring few-shot robustness in out-of-distribution scenarios [26].

Another category of research on policy diversity in RL considers diversity as an independent objective distinguished from reward. An emerging topic called quality-diversity RL aims at learning a set of policies that cover a task-specified behavior space as completely as possible, and maximizing the quality of each policy at the same time [17, 51, 52]. RL is also applied in generative tasks, including code generation, music generation, artificial intelligence for science, and game content generation [1]. Some of these generative tasks treat diversity as an essential need besides quality. For example, Popova et al. formulate drug design as a Markov decision process (MDP) rewarded by a predictive

model to generate diverse new chemical structures [35]. Wang et al. leverage diversity-driven ensemble RL to generate diverse game levels with high quality [50].

**Learning Multimodal Policies** An early attempt in online multimodal RL is soft Q-learning (SQL) [13]. It proposes the maximum-entropy RL framework with an amortized actor. As the actor loss in maximum-entropy RL is a KL-divergence, SQL leverages SVGD to estimate the gradient of the actor loss. Stein soft actor-critic ($S^2AC$) [33] views the entropy-regularized critic as an energy that directly samples actions based on it via SVGD. To improve inference speed, $S^2AC$ proposes to train an additional amortized actor that mimics the SVGD sampler. However, our experiments show that SQL performs poorly in many benchmarks, including MuJoCo, while training $S^2AC$ is much more memory-hungry and slower. Huang et al. propose a model-based RL method learning through evidence lower bound to train multimodal actors [20]. To our best knowledge, a fast and effective algorithm to train an amortized actor was absent prior to our work.

Another line of multimodal RL builds upon diffusion actors. There has been a lot of work investigating offline RL based on diffusion actor [58, 49, 24, 4, 3], but online methods have been a few. Specifically, Yang et al. [53] apply approximated policy improvement to generate target actions and train the diffusion actor to match the target actions. Psenka et al. [36] train the diffusion actor by matching the score-based structure with the action gradient of the Q-function. Ren et al. model the diffusion process as a low-level MDP and train the diffusion actor through a two-stage policy gradient [37]. Consistency policy methods have been demonstrated feasible to train diffusion actors online [7, 5]. Wang et al. propose DACER, an online RL algorithm with entropy regulator to train diffusion actor [48], taking decision diversity into consideration by automatically tuning the scale of an additional action noise. In addition, Ying et al. pre-train a Gaussian actor with intrinsic reward [56] to collect trajectories, and then fine-tune a diffusion actor in downstream tasks. Li et al. cluster the trajectories explored by learning with intrinsic rewards to discover modes, then utilize behavior cloning to train a diffusion actor [28]. Jain et al. propose an energy-based training method for diffusion actors [22]. Ishfaq et al. employ a diffusion model to generate synthetic training samples [21].

There has also been some work on leveraging tractable multimodal policy models [38, 50]. Mazoure et al. leverage normalizing flow to improve exploration [32]. Wang et al. represent decision distribution by Gaussian mixture model to generate diverse game levels [50]. However, these models may be weaker in expressivity compared with intractable models.

## 3 Preliminaries

### 3.1 Deep Reinforcement Learning

Deep reinforcement learning [41] aims at optimizing a policy $\pi$ powered by deep learning to solve an MDP. An MDP involves a state space $\mathcal{S}$, an action space $\mathcal{A}$, a reward function $r(\cdot, \cdot) : \mathcal{S} \times \mathcal{A} \mapsto \mathbb{R}$, and a transition dynamics modeling the transition probability from one state to another given an action. Specifically, this paper focuses on the continuous action space. At each time step $t$, the policy observes a state $S_t \in \mathcal{S}$ to determine a decision distribution $\pi(\cdot|S_t)$ and draw an action $A_t \in \mathcal{A}$ from $\pi(\cdot|S_t)$, and then receives a reward signal $R_t = r(S_t, A_t)$. The objective of RL is optimizing $\pi$ to maximize the expected return $J(\pi) = \mathbb{E}_\pi[\sum_{t=0}^\infty \gamma^t R_t]$. A state-action value function conditioned by a policy $\pi$ is defined as $Q^\pi(s, a) = \mathbb{E}_\pi[\sum_{k=0}^\infty \gamma^k R_{t+k}|S_t = s, A_t = a]$ to evaluate a state-action pair. Prevalent continuous deep RL algorithms [29, 39, 11, 14] primarily follow the actor-critic architecture. In this framework, an actor $\pi_\theta$ models the policy, while a critic $Q_\phi$ learns an approximation of the Q-function conditioned by $\pi_\theta$. The actor is typically optimized by policy gradient [40, 42] based on Q-value approximated by the critic.

### 3.2 Policy Regularization

Maximum entropy RL [59, 44, 45] is a classic framework to optimize both expected return and diversity. Haarnoja et al. introduce a deep learning framework for entropy-regularized RL [14], which modifies the standard RL objective by adding an entropy regularization $J(\pi) = \mathbb{E}_\pi[\sum_{t=0}^\infty \gamma^t (R_t + \alpha \mathcal{H}(\pi(\cdot|S_t)))]$, where $\mathcal{H}(\cdot)$ indicates the entropy of a given distribution. The state-action value function is defined as $Q_{\text{soft}}^\pi(s, a) = \mathbb{E}[R_t + \sum_{k=1}^\infty \gamma^k (R_{t+k} + \alpha \mathcal{H}(\pi(\cdot|s)))|S_t = s, A_t = a]$. The

policy is optimized by applying a soft policy iteration (SPI) operator as follows

$$\pi_{\text{new}} \leftarrow \underset{\pi}{\arg\min} \, D_{\text{KL}} \left( \pi(\cdot|s) \, \bigg\| \, \frac{\exp(Q^{\pi_{\text{old}}}_{\text{soft}}(s, \cdot))}{Z^{\pi_{\text{old}}}(s)} \right), \tag{1}$$

where $Z^{\pi_{\text{old}}}(s)$ is a scalar to normalize the distribution, which is intractable but negligible in practice.

Wang et al. extend the entropy-regularized RL framework to a general form [50]. Given an arbitrary regularization function $\varrho(\pi(\cdot|s))$, and a regularized learning objective $J(\pi) = \mathbb{E}_\pi[\sum_{t=0}^{\infty} \gamma^t (R_t + \alpha \varrho(\pi(\cdot|S_t)))]$, the regularized Q-value is defined in the following form

$$Q^{\pi}_{\text{reg}}(s, a) = \mathbb{E}_\pi \left[ R_t + \sum_{k=1}^{\infty} \gamma^k (R_{t+k} + \alpha \varrho(\pi(\cdot|S_{t+k}))) \, \bigg| \, S_t = s, A_t = a \right]. \tag{2}$$

With the regularized Q-value defined, the policy can be optimized by regularized policy iteration with a policy improvement operator expressed as

$$\pi_{\text{new}}(\cdot|s) \leftarrow \underset{\pi}{\arg\max} \left[ \alpha \varrho(\pi(\cdot|s)) + \mathbb{E}_{a \sim \pi(\cdot|s)} [Q^{\pi_{\text{old}}}_{\text{reg}}(s, a)] \right] \tag{3}$$

for all $s \in \mathcal{S}$. Furthermore, the policy gradient of the regularized learning objective is

$$\nabla_\theta J(\pi) = \mathbb{E}_{s \sim d^\pi} \left[ \alpha \nabla_\theta \varrho(\pi_\theta(\cdot|s)) + \int_{\mathcal{A}} Q^{\pi}_{\text{reg}}(s, a) \nabla_\theta \pi_\theta(a|s) \, \mathrm{d}a \right], \tag{4}$$

where $d^\pi$ is the discounted state distribution given $\pi$.

## 4 Method

We propose diversity-regularized actor-critic (DrAC), which learn intractable multimodal policies through reparameterization and diversity regularization. Fig. 2 illustrates the key ingredients of DrAC. A detailed pseudo-code is provided in Appendix A.1.

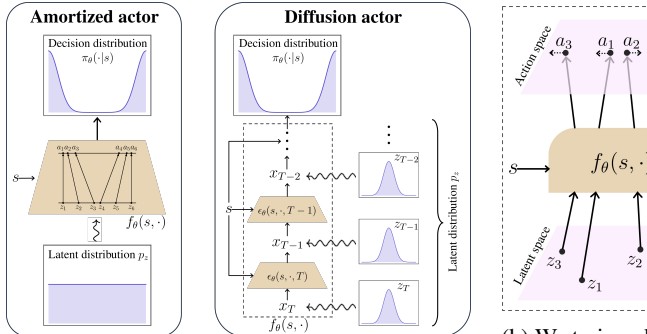

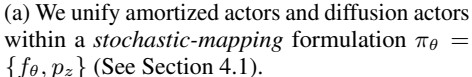

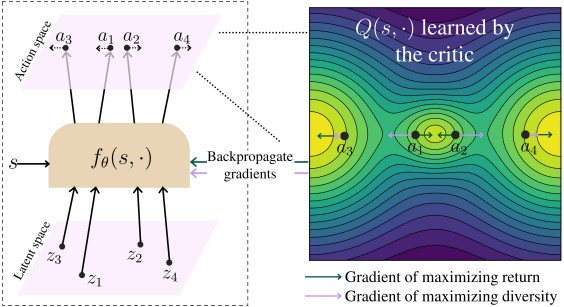

(a) We unify amortized actors and diffusion actors within a *stochastic-mapping* formulation $\pi_\theta = \{f_\theta, p_z\}$ (See Section 4.1).

(b) We train a diversity-regularized critic and directly backpropagate the gradient of $Q$ and diversity to the actor's weights, as $\nabla_\theta Q(s, f_\theta(s, z))$ is an estimator of policy gradient via reparameterization trick.

Figure 2: Key ingredients of our proposed method.

### 4.1 Unifying Intractable Multimodal Policies and Policy Gradient

We define a class of *stochastic-mapping actors* as $\pi_\theta = \{f_\theta, p_z\}$, which is a combination of a parameterized function $f_\theta(\cdot, \cdot) : \mathcal{S} \times \mathcal{Z} \mapsto \mathcal{A}$, and a fixed latent distribution $p_z$ over a latent space $\mathcal{Z}$. The action is drawn by sampling an independent random variable $z$ from $p_z$ first, and then feeding state $s$ and $z$ into $f_\theta$, i.e., $a \sim \pi_\theta(\cdot|s) \equiv a \leftarrow f_\theta(s, z), z \sim p_z$. We reformulate existing multimodal actors, namely *amortized actors* and *diffusion actors*, under this definition as follows.

**Amortized actors**   [2] This type of actors [13] employ a neural network (NN) $g_\theta$, which takes the state $s$ and the latent variable $z$ as input, and directly output the action $a$. We follow the most straightforward mechanism in SQL, concatenate $s$ and $z$ directly feed into $g_\theta$. Formally, this is

$$f^{\text{Amort}}_\theta(s, z) \equiv g_\theta(s \oplus z). \tag{5}$$

---

[2]This definition follows [13] and [33]. It is narrower than the concept of amortized inference.

**Diffusion actors**   This type of actors [36, 37, 48] build upon the diffusion models [18], which are powerful generative models. We adopt the implementation from DACER [48]. The policy is parameterized by an NN $\epsilon_\theta(\cdot, \cdot, \cdot)$, and can be formulated by $\pi_\theta = \{f_\theta, p_z\}$ as follows

$$f_\theta^{\text{Diffus}}(s, z) \equiv x_0,$$
$$x_{t-1} = \frac{1}{\sqrt{\alpha_t}} \left( x_t - \frac{\beta_t}{\sqrt{1 - \bar{\alpha}_t}} \epsilon_\theta(s, x_t, t) \right) + \sqrt{\beta_t} z_{t-1}, \tag{6}$$
$$x_T = z_T.$$

In Eq. (6), $\alpha_t$ and $\beta_t$ are scalars calculated by some specific schedule mechanisms to control the diffusion process, $z_0, z_1, \cdots, z_T$ are i.i.d. noise vectors drawn from $\mathcal{N}(\mathbf{0}, I)$ and we have treated $z = \{z_0, \cdots, z_T\}$, $T$ is the number of diffusion steps. Note that the $t$'s here are not the time steps of MDP but the time steps of the diffusion process.

These actors are powerful, but their decision probabilities $\pi_\theta(a|s)$ does not have a closed-form expression, making policy gradient not directly applicable. However, as they follow the stochastic mapping formulation, it is feasible to calculate policy gradient via reparameterization trick (PGRT) as mentioned in [27] by the equation below

$$\nabla_\theta J(\pi_\theta) = \mathbb{E}_{s \sim d^\pi, z \sim p_z}[\nabla_a Q(s, f_\theta(s, z)) \nabla_\theta f_\theta(s, z)]. \tag{7}$$

PGRT demonstrates that we can directly backpropagate the Q-function's gradient to $f_\theta$ to train stochastic-mapping actors, where the Q-function can be approximated by differentiable critics. We noticed that DACER [48] backpropagates the gradient of the Q-function to train diffusion actors, but the paper of DACER did not mention the relation to policy gradient and did not disclose that this method can be generalized to other actors.

## 4.2   Actor-Critic Learning with Diversity Regularization

As we pursue decision diversity, an effective diversity regularization is desired. As discussed earlier, the traditional entropy regularization do not prefer multimodality and is not applicable to intractable actors. To overcome this challenge, we propose a distance-based regularization to encourage multimodal diversity.

A straightforward distance-based diversity metric is the mean pairwise distance, formulated as $D^\pi(s) = \mathbb{E}_{x,y \sim \pi(\cdot|s)}[\delta(x, y)]$, where $\delta(\cdot, \cdot)$ is a distance metric. However, this metric may overestimate the diversity when the data distribution forms some tiny clusters far away from each other (See Fig. 3). We empirically found that this issue led to underwhelming performance. Therefore, we consider the geometric mean $\text{GM}(\{\delta(x_1, y_1), \cdots, \delta(x_n, y_n)\}) = \left(\prod_{i=1}^n \delta(x_i, y_i)\right)^{-1/n}$ of pairwise distances instead. The geometric mean is sensitive to small values as multiplication is taken instead of summation, which mitigates the overestimation issue (See Fig. 3). Furthermore, we reshape diversity on the log scale to make it easier to balance reward and diversity. We propose a diversity regularization as follows

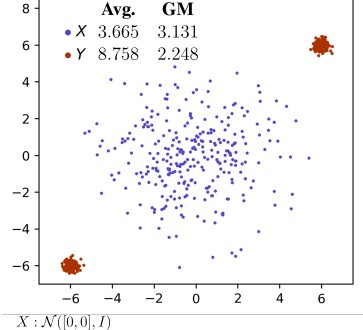

$X : \mathcal{N}([0, 0], I)$

$Y$ : A mixture of $\mathcal{N}([6, 6], 0.2I)$ and $\mathcal{N}([-6, -6], 0.2I)$

Figure 3: Average (Avg.) and geometric mean (GM) of pairwise distances for two synthesized data distributions. Average pairwise distance overestimates the diversity of distribution $Y$.

$$D^\pi(s) = \mathbb{E}_{x \sim \pi(\cdot|s), y \sim \pi(\cdot|s)}[\log \delta(x, y)]. \tag{8}$$

Specifically, L2 distance is adopted as the distance metric. This diversity regularization is added to the original RL objective with a coefficient $\alpha$, resulting in an objective to maximize as

$$J = \mathbb{E}_\pi \left[ \sum_{t=0}^\infty \gamma^t (R_t + \alpha D^\pi(S_t)) \right]. \tag{9}$$

We estimate the diversity of the actor $\pi_\theta$ at a certain state $s$ by sampling $n$ pairs of i.i.d. latent vectors $(z_1^x, z_1^y), \cdots, (z_n^x, z_n^y)$, and compute the average log-distance over samples. This is denoted as $\tilde{D}_\theta(s) = \frac{1}{n} \sum_{i=1}^n \log \delta(f_\theta(s, z_i^x), f_\theta(s, z_i^y))$. Though this estimation requires multiple samples, it is time-efficient as such a process can be easily parallelized on GPUs.

The practical DrAC algorithms follow the common actor-critic architecture. The actor is a stochastic-mapping actor, and the critics follow the learning tricks in soft actor-critic (SAC) [14]. Double critics $Q(\cdot, \cdot; \phi_1), Q(\cdot, \cdot; \phi_2)$ and double target critics $\hat{Q}(\cdot, \cdot; \hat{\phi}_1), \hat{Q}(\cdot, \cdot; \hat{\phi}_2)$ are employed. By substituting $\tilde{D}_\theta$ into Eq. (2) and unrolling it into a bootstrap learning target, the critic loss is derived as

$$\mathcal{L}_\phi = \mathbb{E}_{s,a,r,s'\sim\mathcal{D}}\left[\text{MSE}(Q(s,a;\phi_i), r + \gamma(\tilde{V}(s';\hat{\phi}) + \alpha\tilde{D}_\theta(s')))\right], i \in \{1,2\} \tag{10}$$

where $\mathcal{D}$ represents the off-policy replay buffer, $\tilde{V}(s';\hat{\phi}) = \min_{i\in\{1,2\}}\hat{Q}(s',a';\hat{\phi}_i)$ is an approximation of the value of the next state $s'$ based on the target critics and $a'$ is a sample drawn from $\pi_\theta(\cdot|s')$. The target critics are updated by the target smoothing strategy [14, 11].

Regarding actor learning, we combine PGRT (See Eq. (7)) with the regularized policy gradient theorem (See Eq. (4)), derive a loss function as follows

$$\mathcal{L}_\theta = -\mathbb{E}_{s\sim\mathcal{D},z\sim p_z}[Q(s, f_\theta(s,z); \phi) + \alpha\tilde{D}_\theta(s)], \tag{11}$$

where $Q(s, f_\theta(s,z); \phi) = \min_{i\in\{1,2\}}Q(s, f_\theta(s,z); \phi_i)$, following the technique used in SAC. This loss function can also be considered an approximation of Eq. (3).

### 4.3 Automatic Coefficient Adjustment based on Target-Diversity

Multimodal actors are expressive and flexible. However, given the diversity regularization with a fixed coefficient, such properties can also lead to instability in critic learning, which may harm the training. To mitigate this issue and also make hyperparameter tuning easier, we borrow the automatic coefficient adjustment technique from SAC [15]. Specifically, we parameterize $\alpha$ with a learnable scalar $w$ such that $\alpha = \exp(w)$ if $w < 0$ and $\alpha = w + 1$ otherwise. Then $\alpha$ is treated as learnable and updated by descending the gradient of a dual-optimization loss defined as

$$\mathcal{L}_\alpha = \mathbb{E}_{s\sim\mathcal{D}}[\alpha(\tilde{D}_\theta(s) - \hat{D})], \tag{12}$$

where $\hat{D}$ is a target diversity value. Minimizing this loss will make $\tilde{D}_\theta(s)$ match with $\hat{D}$ adaptively. As $\tilde{D}_\theta(s)$ is defined on the log-scale, determining its range can be trivial. We introduce a temperature hyperparameter $\beta$ such that $\hat{D} \leftarrow \log(\beta\sqrt{|\mathcal{A}|})$ to ease the triviality. The scale of $\beta$ is comparable to pairwise distances after normalization by the action space dimensionality $|\mathcal{A}|$.

## 5 Experiments

**Domains**  Previous works on multimodal RL [13, 33, 48] only verify the multimodal ability with a toy 2D multi-goal navigation task and mainly focus on the improvements in MuJoCo. To demonstrate the advantages of DrAC and multimodal policies, we include two diversity-critical benchmarks in *multi-goal achieving* and *generative RL* domains. We also test DrAC and representative multimodal RL algorithms in standard MuJoCo benchmarks to compare the general performance.

**Baselines**  By installing DrAC with an amortized actor and a diffusion actor separately, we obtain two versions of DrAC, namely DrAmort and DrDiffus. We include SQL [13], DACER [48], $S^2$AC [33] as multimodal RL baselines, and SAC [15] as a unimodal baseline. Table 1 summarizes the techniques of these algorithms. We keep NN architectures and common hyperparameters all the same as the default setting of SAC to enable a fair comparison [14]. Appendix A.2 details the hyperparameters. All trainings are conducted with five seeds.

### 5.1 Multi-goal Achieving

In real-world scenarios, there could be multiple goals to achieve. We build a multi-goal version of the PointMaze environment in D4RL [10] to conduct a case study of multi-goal achieving. This environment requires the agent to learn to steer a ball from a fixed original point to navigate to one of the goals within a maze. The maze contains multiple goals. The reward is sparse, only reaching a goal returns $+100$ reward and ends the episode with a success flag. Besides the success rate, the number of reachable goals is also evaluated. We designed three maze maps with increasing difficulty levels, as illustrated in Fig. 9. We notice that the work of [28] tested more complex multi-goal achieving

Table 1: A summary of compared algorithms. Bold texts indicate our contributions. The "Temperature" column indicates the technique of adjusting the emphasis on diversity used by each algorithm.

|  | Actor Model | Actor Learning | Critic Learning | Diversification | Temperature |
|---|---|---|---|---|---|
| **DrAmort** | Amortized | **Regularized PGRT** | Conventional | **Max-Diversity** | Target-Matching |
| SQL [13] | Amortized | SPI via SVGD | Conventional | Max-Entropy | Constant |
| **DrDiffus** | Diffusion | **Regularized PGRT** | Conventional | **Max-Diversity** | Target-Matching |
| DACER [48] | Diffusion | PGRT | Distributional [8] | Noise Scaling | Target-Matching |
| $S^2AC$ [33] | Energy-based | SPI via SVGD | Conventional | Max-Entropy | Constant |
| SAC [14] | Gaussian | SPI | Conventional | Max-Entropy | Target-Matching |

benchmarks. However, that work autonomously collects and clusters trajectories first, then clones behaviors using these clusters, while our work focuses on basic RL algorithms. To ensure a fair and meaningful comparison, we conducted a grid search for each algorithm before the formal evaluation. The highest temperature that ensures each algorithm reaches the optimal success rate is picked.

The trajectories in the simple maze of learned policies are visualized in Fig. 4. It is clear that DrAmort learns the most diverse and uniform trajectories. Despite using the same diffusion actor, DrDiffus learn to reach different goals, DACER does not. Fig. 5b shows that DrAmort and SQL consistently learns to reach the most goals in all mazes, and DrAmort learns faster than SQL. In terms of success rate, DrAmort, DrDiffus, and DACER learn fast and stably, while SQL, SAC, and $S^2AC$ exhibit a decrease in success rate in the early stage and reach optimal success rate more slowly. Given the results, we consider that amortized actors possess superior multimodal expressivity. DACER struggles with learning multimodal behaviors in PointMaze. This is most likely due to DACER only controlling the diversity via noise scaling. Meanwhile, SQL shows better multimodality than DrDiffus. The underwhelming multimodality of diffusion actors may be owing to a small number of diffusion steps of 20 we used as suggested in [48]. However, despite using a small number of diffusion steps, the inference and training speed is much slower than amortized actors. Therefore, we advocate that the amortized actor is a promising actor class for representing multimodal policies. Trajectories in medium and hard mazes are presented in Appendix B.2.

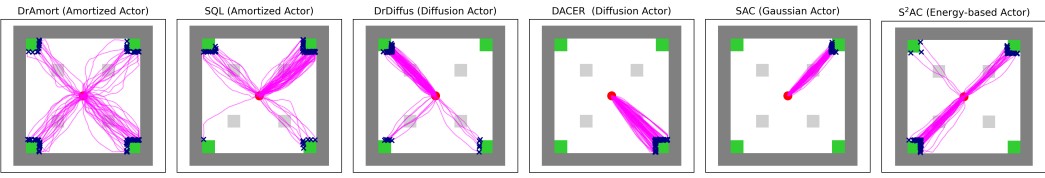

Figure 4: Evaluation trajectories of all tested algorithms in the simple maze.

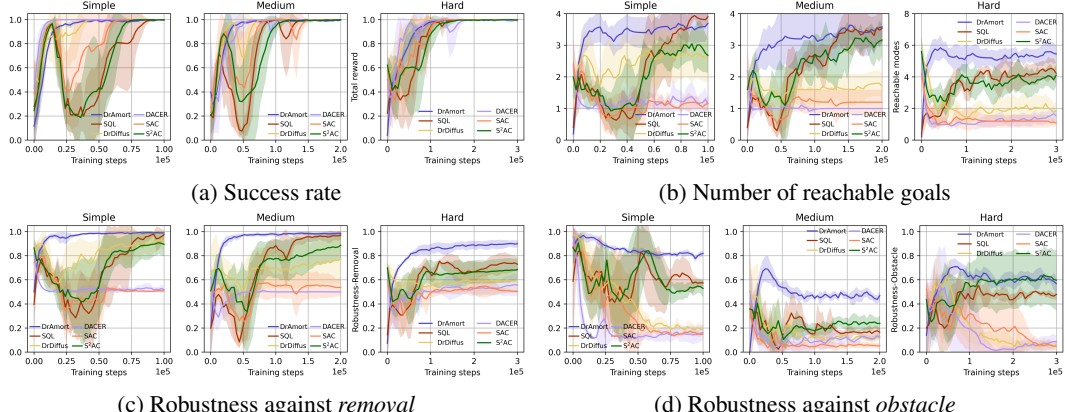

(a) Success rate

(b) Number of reachable goals

(c) Robustness against *removal*

(d) Robustness against *obstacle*

Figure 5: Learning curves in multi-goal PointMaze. Solid lines represent mean performance, and shaded regions indicate standard deviation. All curves are smoothed by the exponential moving average with a coefficient of $0.5$.

Inspired by [26], we investigate whether diversity empowers few-shot robustness in out-of-distribution scenarios. We consider two types of perturbations: 1) *Removal*: half of the goals are removed during the test; 2) *Obstacle*: some obstacles that do not exist in the training phase, appear in the test phase and block the shortest paths to the goals. The first case requires agents to reach all goals with uniform frequency. The second case requires the agents to navigate to the goal with diverse trajectories, so that the they can bypass the obstacles by chance. Similar to [26], we test the *five-episode success rate* of all algorithms in each scenario. That is, the probability that the agent can successfully reach any goal at least once within five independent trials. The results are shown in Fig. 5c and Fig. 5d.

DrAmort shows the best robustness in all mazes, while SQL outperforms others among the rest algorithms. According to Fig. 4, DrAmort exhibits the best trajectory diversity, so it has more chances to bypass the obstacles. Meanwhile, DrAmort reaches goals more uniformly, so higher robustness against removal is guaranteed. The results confirm that multimodality and diversity empower few-shot robustness in out-of-distribution scenarios. DrDiffus outperforms DACER regarding robustness against removal. Regarding SAC, although a high temperature is set, it struggles in learning to reach multiple goals and generalizing to out-of-distribution test cases. This phenomenon confirms the fundamental limitation of unimodal actors.

## 5.2 Generative RL

There has been some application of RL in generative tasks, e.g., game content generation [25, 50] and program generation [30]. We take game level generation benchmark in [50] as a case study because policy multimodality is important to balance quality and diversity in this domain, according to [50].

In this game level generation benchmark, the agent needs to observe a sliding window on an existing game level and propose new level segments to be concatenated with the existing level. The level segment is represented by a decoder, which is the generator of pretrained generative adversarial networks [12, 47]. That means the agent will output a continuous latent vector at each step, and the decoder will decode the action into a piece of level. There are two expert knowledge-based formulations of the reward function in this benchmark, which aim to evaluate the quality of generated levels under some game design principles. The reward also includes a penalty for unsolvable level segments to guide agents to generate solvable levels. Solvable or not is checked by rules in our experiments. Different reward functions result in different styles of generated levels, namely *MarioPuzzle* and *MultiFacet*. Fig. 6 shows example levels of the two styles generated by DrAmort. Besides episodic return, the diversity of generated levels is also desired in this domain.

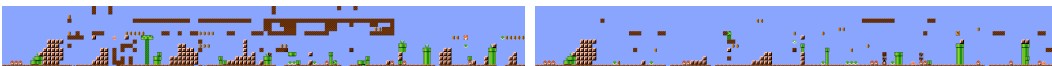

Figure 6: Example *MarioPuzzle* style (left) and *MultiFacet* style (right) levels generated by DrAmort.

We train algorithms with the same temperature settings as multi-goal PointMaze. The learning curves are shown in Fig. 7a. For both styles, DrAmort shows the best episodic returns, namely the best qualities. This is possibly attributed to the superior expressivity of the amortized actor. As the distribution of high-quality content is most likely multimodal, only highly expressive actors can represent high-performance policies under high temperatures. Though SQL also applies amortized actor, it performs badly. As SQL directly uses SVGD as the gradient estimator for approximating SPI, it is possible that reparameterization trick is a better gradient estimator than SVGD for RL. $S^2AC$, which applies SVGD to directly sample actions, performs the worst on all tasks. On the other hand, DrDiffus and DACER do not outperform SAC.

We also evaluate the diversity of generated levels by computing the average pairwise Hamming distance following [50], and then visualize the performance in the 2D quality-diversity objective space of all trained policies in Fig. 7b. Each marker in the figure represents a policy trained by one of the algorithms with an independent seed. Under MarioPuzzle style, DrAmort generally dominates other algorithms, i.e., outperforms other algorithms in terms of both return and diversity. Under the MultiFacet style, DrAmort dominates other algorithms except for SQL. Though SQL shows the best diversity, its episodic return is poor. Under the other style, SQL does not demonstrate better diversity, and the episodic return is still poor. DrDiffus, DACER, and SAC are non-dominated with each other. But diffusion actor is much slower than Gaussian and amortized actors. Therefore, diffusion

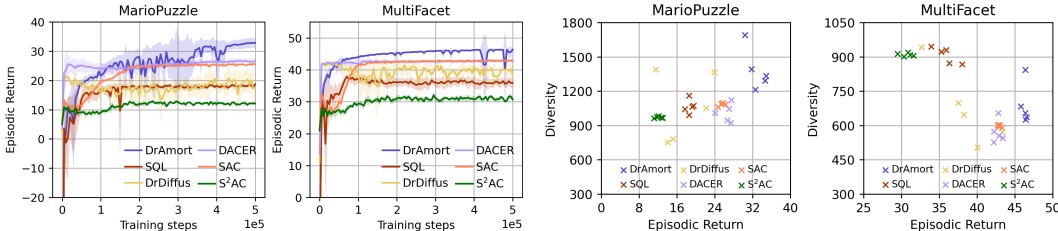

(a) Learning curves regarding average episodic return, i.e., average quality of generated levels.

(b) Locations of all trained policies in the episodic return-diversity performance space.

Figure 7: Evaluation results in the game content generation benchmark.

actors might not be a good choice for this task. We present t-SNE embedding visualizations [46] of generated levels in Appendix B.5, to demonstrate their diversity more intuitively.

## 5.3 Conventional Benchmarks

To evaluate the general performance of DrAC in conventional tasks, we test DrAC in MuJoCo-v4 and compare it with baselines. We do not include $S^2AC$ as it is too computationally intensive and it uses an unusual implicit actor, making it less meaningful to compare standard performance. Learning curves in the six MuJoCo locomotion tasks are presented in Fig. 8.

DrAmort exhibits competitive performance. It achieves the best performance in three out of the six tasks and surpasses SQL in a total of five, demonstrating that DrAC is a strong method for training amortized actors. Overall, the performance of DrDiffus is comparable to DACER, though DACER leverages a more advanced critic learning technique [8]. Therefore, DrAC can serve as a high-performance base algorithm for learning multimodal policies, particularly for amortized actors. Meanwhile, it is surprising that DACER and DrDiffus

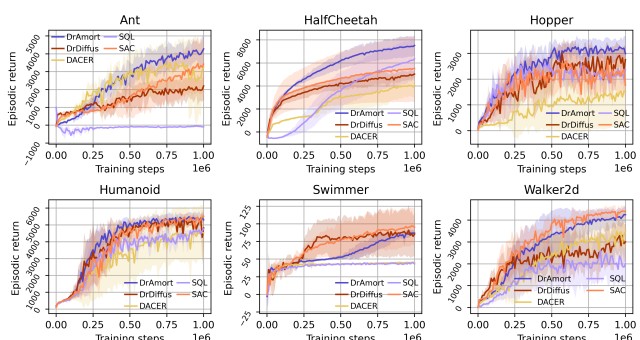

Figure 8: Learning curves in MuJoCo.

do not outperform SAC, though they apply more complex diffusion actors. We notice that the paper of DACER [48] uses deeper networks, more advanced activations, and smaller learning rates than the default settings of [14]. It is possible that diffusion actors need more powerful networks and more elaborate hyperparameter tuning to fulfill their potentials.

## 6 Conclusion

We propose _Diversity-regularized Actor Critic_ (DrAC), a novel and versatile RL algorithm designed to train intractable multimodal policies. DrAC addresses the intractability challenge in training multimodal policies by leveraging the reparameterization trick and distance-based diversity regularization. Meanwhile, DrAC is compatible with all policy models that can be formulated as a stochastic-mapping actor defined as $\pi_\theta = \{f_\theta, p_z\}$ such that $a \sim \pi_\theta(\cdot|s) \equiv a \leftarrow f_\theta(s, z), z \sim p_z$. Experiments show DrAC can train amortized actors to achieve superior performance in multi-goal achieving and generative RL benchmark, which also verifies the significance of multimodal policies in these domains. Notably, using an amortized actor, DrAC demonstrates strong few-shot robustness [26] in out-of-distribution test scenarios within the multi-goal PointMaze environment. Nevertheless, this paper only investigates a basic implementation of DrAC, leaving ample room for further improvements. Future work could investigate scheduling the temperature, designing better stochastic-mapping actors, exploiting other distance and diversity metrics, and integrating more learning tricks, etc.

Our empirical results also indicate that the amortized actor possesses strong multimodal expressivity and performs well in all domains involved in this paper. On the other hand, the diffusion actor does not show significant advantages. It is possible that diffusion actors need more careful tuning of hyperparameters and better NN architecture design. However, the training and inference speed of amortized actors is comparable to traditional actors and significantly faster than diffusion actors. Therefore, we posit that the amortized actor represents a promising actor class for multimodal RL.

## Acknowledgments and Disclosure of Funding

This work is supported by the National Natural Science Foundation of China 62406266.

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

# A Technical Details

## A.1 Pseudo code of DrAC

Algorithm 1 details DrAC.

---

**Algorithm 1** Diversity-regularized Actor-Critic (DrAC)

---

**Require:** Actor weights $\theta$; Critic weights $\phi = \{\phi_1, \phi_2\}$; Target critic weights $\hat{\phi} = \{\hat{\phi}_1, \hat{\phi}_2\}$
  Let gradient_step_count $\leftarrow 0$
  **for** $t : 1 \rightarrow T$ **do**
    Sample an action $a \sim \pi(\cdot|s)$
    Execute $a$ and observe $s', r, d$
    $r, s', d \leftarrow$ Execute $a$ in the environment
    Store $(s, a, r, s, d)$ in the buffer
    **while** gradient_step_count $< t * $ reply_ratio **do**
      Sample a batch $(s, a, r, s', d)$ from the replay memory
      Sample $z$; $(z_1^x, z_1^y), \cdots, (z_n^x, z_n^y)$ from $p_z$
      Let $\tilde{D}_\theta(s') \leftarrow \frac{1}{n} \sum_{i=1}^n \log \delta(f_\theta(s', z_i^x), f_\theta(s', z_i^y))$
      Compute Q-target $y \leftarrow r + \gamma(1-d)(\hat{Q}(s, f_\theta(s, z); \hat{\phi}) + \alpha \tilde{D}_\theta(s'))$
      Update $\phi_1$ and $\phi_2$ with MSE$(Q(s, a; \phi_i), y), i \in \{1, 2\}$
      Sample $z$; $(z_1^x, z_1^y), \cdots, (z_n^x, z_n^y)$ from $p_z$
      Let $\tilde{D}_\theta(s) \leftarrow \frac{1}{n} \sum_{i=1}^n \log \delta(f_\theta(s, z_i^x), f_\theta(s, z_i^y))$
      Update $\theta$ with $\mathcal{L}_\theta = -Q(s, f_\theta(s, z); \phi) - \alpha \tilde{D}_\theta(s))$
      Update $\alpha$ with $\mathcal{L}_\alpha = \alpha(\tilde{D}_\theta(s) - \hat{D})$
      Update $\hat{\phi}$ by $\hat{\phi} \leftarrow \rho\phi + (1-\rho)\hat{\phi}$
      Let gradient_step_count $\leftarrow$ gradient_step_count $+ 1$
    **end while**
  **end for**

---

## A.2 Experiment Settings

The customized maze maps used in our multi-goal PointMaze environment are shown in Fig. 9

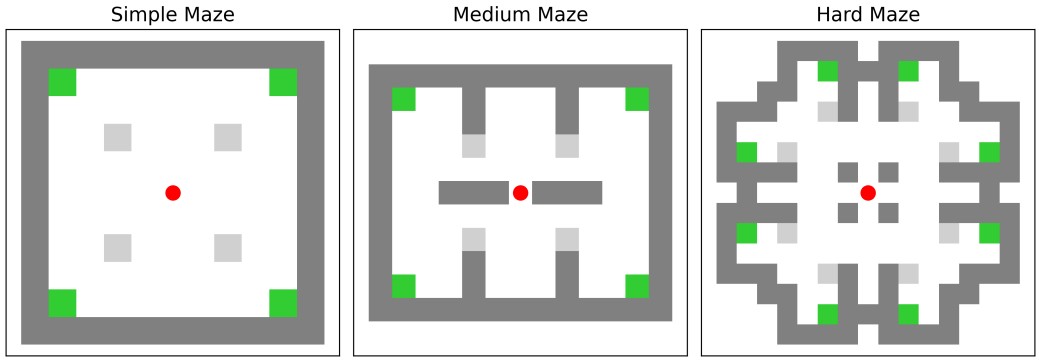

Figure 9: Maze maps of the Multi-goal PointMaze environment. The red circle indicates the start point, green cells indicate goals, gray cells are walls, and light gray cells are obstacles that only appear in the robustness evaluation stage.

The hyperparameters are listed in Table 2. The temperature hyperparameters listed in Table 2 are used in the multi-goal PointMaze environment and the game content generation environment. While for MuJoCo, we use a lower temperature for DrAC. For baseline algorithms, we use the default temperatures MuJoCo provided in their original papers. Temperature hyperparameters for all algorithms in MuJoCo are listed in Table 3.

Table 2: Hyperparameters

| Hyperparameter | **DrAmort** | **DrDiffus** | DACER | SQL | S$^2$AC | SAC |
|---|---|---|---|---|---|---|
| Replay buffer size | 1000000 | | | | | |
| MLP hidden layers in actors & critics | (256, 256) | | | | | |
| Hidden layer activation in all networks | ReLU | | | | | |
| Optimizer | Adam | | | | | |
| Learning rate of actors & critics | $3e^{-4}$ | | | | | |
| Discount rate ($\gamma$) | 0.99 | | | | | |
| Target smoothing coefficient ($\rho$) | 0.005 | | | | | |
| Replay ratio (gradient step) | 1 | | | | | |
| Mini-batch size | 256 | | | | | |
| Diversity temperature ($\beta$) | 0.8 | | N/A | | | |
| Target entropy ($\bar{\mathcal{H}}$) | N/A | | $0.5\|\mathcal{A}\|$ | N/A | | $0.5\|\mathcal{A}\|$ |
| Learning rate for $\alpha$ auto-adjustment | $5e^{-3}$ | | $3e^{-2}$ | N/A | | $3e^{-4}$ |
| Entropy coefficient ($\alpha$) | N/A | | | 0.3 | 0.7 | N/A |
| Number of diversity estimation pairs ($n$) | 8 | | N/A | | | |
| Number of SVGD particles ($k$) | N/A | | | 10 | 32 | N/A |
| Policy delay (actor update period) | N/A, equivalent to 1 | | 2 | N/A, equivalent to 1 | | |

Table 3: Temperature settings for MuJoCo

| **DrAmort** | **DrDiffus** | DACER | SQL | S$^2$AC | SAC |
|---|---|---|---|---|---|
| 0.5 | 0.2 | $-0.9\|\mathcal{A}\|$ [48] | 0.2 for Ant, 1.0 for the rest [33] | | $-\|\mathcal{A}\|$ |

Regarding compute resources, all of our experiments are conducted with a Linux server with 8 NVIDIA RTX 3090 GPUs and an Intel Xeon Platinum 8375C CPU. All trainings are done with one GPU.

### A.3 Relation to Amortized Inference and Traditional Gaussian Actor

The formulation of stochastic mapping actors is closed to the definition of amortized inference. Due to that the term "amortized actor" has been used in [13] and adopted in [33], we name our formulation as "stochastic mapping actors".

The formulation of stochastic mapping actors is also compatible with traditional Gaussian actor, by treating $f(s,z) \equiv \mu_\theta(s) + \sigma_\theta z$ and $z \sim \mathcal{N}(\mathbf{0}, I)$. Our method is also directly applicable to Gaussian actor.

# B   Additional Results

## B.1   Computational costs

We record the average time cost per 1000 training steps in Ant-v4 environment. The test is conducted on a Linux server with 8 NVIDIA RTX 3090 GPUs and an Intel Xeon Platinum 8375C CPU. For each algorithm, we run three trials on GPU cards 0, 1 and 2, respectively, and then average the training time per 1000 steps. All algorithms are implemented with PyTorch 2.5.1, and the CUDA version is 12.5.

Table 4: Computational costs of all compared algorithms

|  | SAC | SQL | **DrAmort** | DACER | **DrDiffus** | S$^2$AC |
|---|---|---|---|---|---|---|
| Training time per 1000 steps | 22.7s | 18.9s | 21.0s | 76.9s | 84.8s | 220.5s |
| Peak GPU memory usage | 352M | 472M | 392M | 364M | 608M | 3462M |

According to the table, we found that DrAmort is even faster than SAC. The reason is that SAC computes and back-propagates the log-probability at certain actions, which requires more sequential tensor operations. Though DrAmort requires samples, the computation is parallelized and takes fewer sequential tensor operations. As a result, DrAmort uses fewer GPU clock ticks and is slightly faster than SAC, given that the neural network is sufficiently small compared to the GPU's parallel computation capability. For DrDiffus, it takes about 10% more time for training compared to DACER, but its diversity in multi-goal PointMaze is better.

For large-scale tasks that require larger networks, such as visual control, we can inject the latent variable at the last few layers. For example, if a CNN is used, the latent variables can be injected after the convolutional layers. In this case, the computation time and memory usage will not significantly increase.

## B.2   Trajectories in Medium and Hard Multi-goal PointMaze

Figs. 10 and 11 demonstrate the trajectories in medium and hard mazes, respectively.

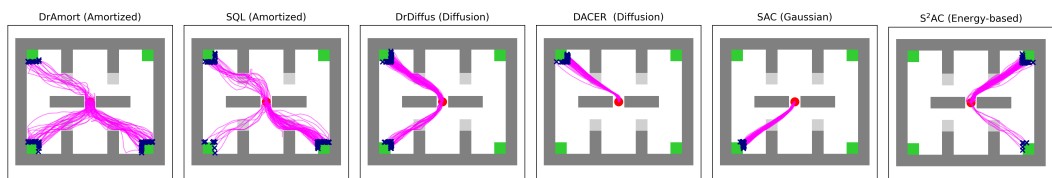

Figure 10: Evaluation trajectories of all tested algorithms in the medium maze.

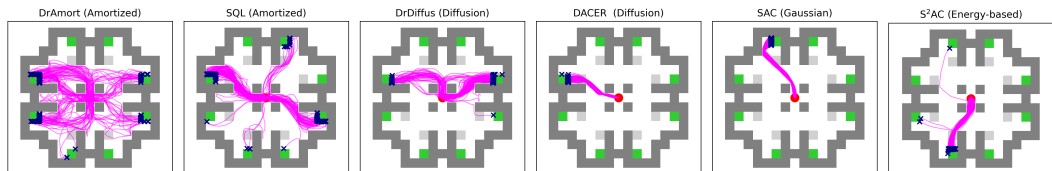

Figure 11: Evaluation trajectories of all tested algorithms in the hard maze.

Consistently, DrAmort exhibits the best diversity.

## B.3 Sensitivity of Temperature Hyperparameter in Multi-goal PointMaze

We display the learning curves of all evaluated algorithms with varied temperatures. For our DrAmort and DrDiffus, we show a sensitivity study with five different temperatures. For baseline algorithms, we present three temperatures to explain our choice of their temperatures in our comparison study. Specifically, for all baseline algorithms, using a higher temperature than our choice leads to suboptimal success rate in at least one maze; while using a lower one leads to lower diversity.

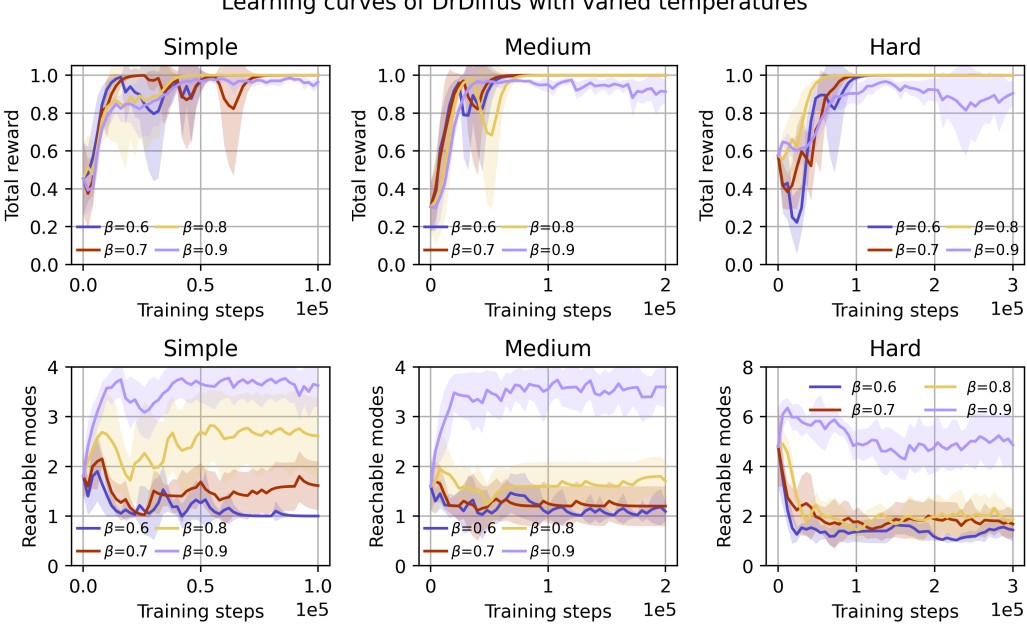

Figure 12: Learning curves of DrAmort in multi-goal PointMaze with varied temperatures.

Figure 13: Learning curves of DrDiffus in multi-goal PointMaze with varied temperatures.

Learning curves of SQL with varied temperatures

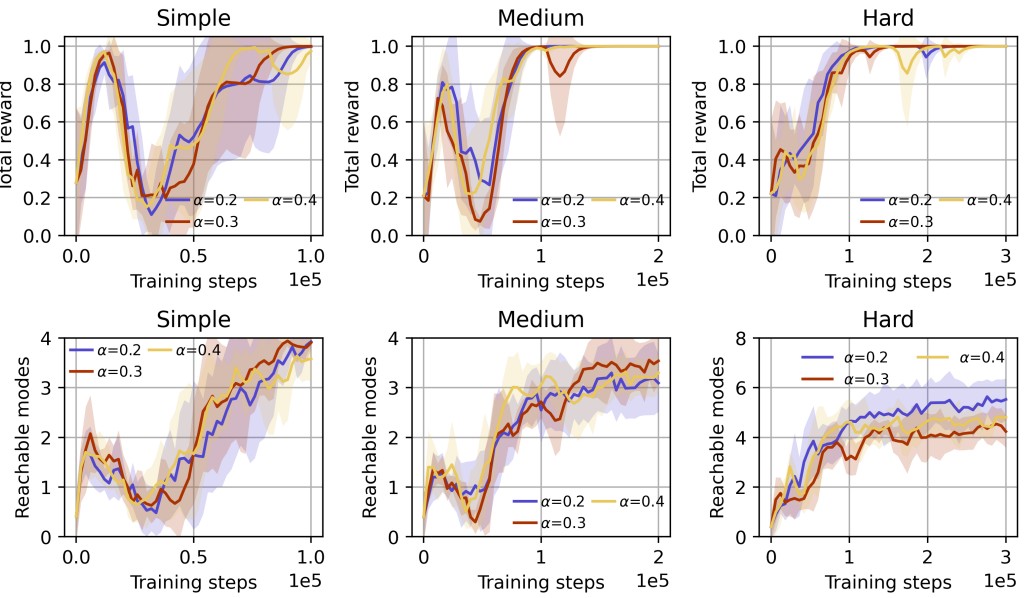

Figure 14: Learning curves of SQL in multi-goal PointMaze with varied temperatures.

Learning curves of DACER with varied temperatures

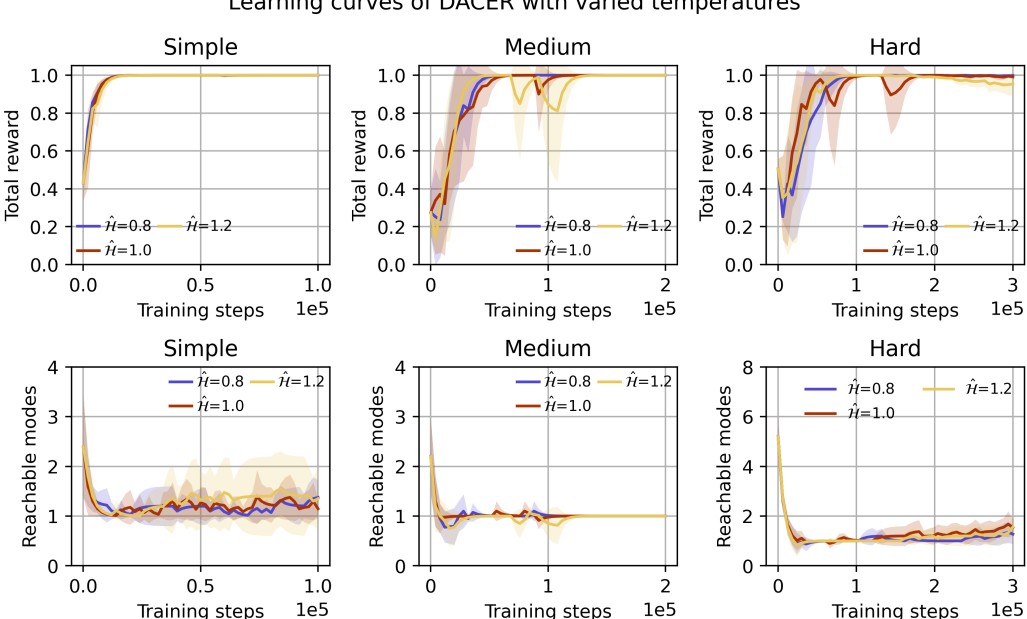

Figure 15: Learning curves of DACER in multi-goal PointMaze with varied temperatures.

Learning curves of SAC with varied temperatures

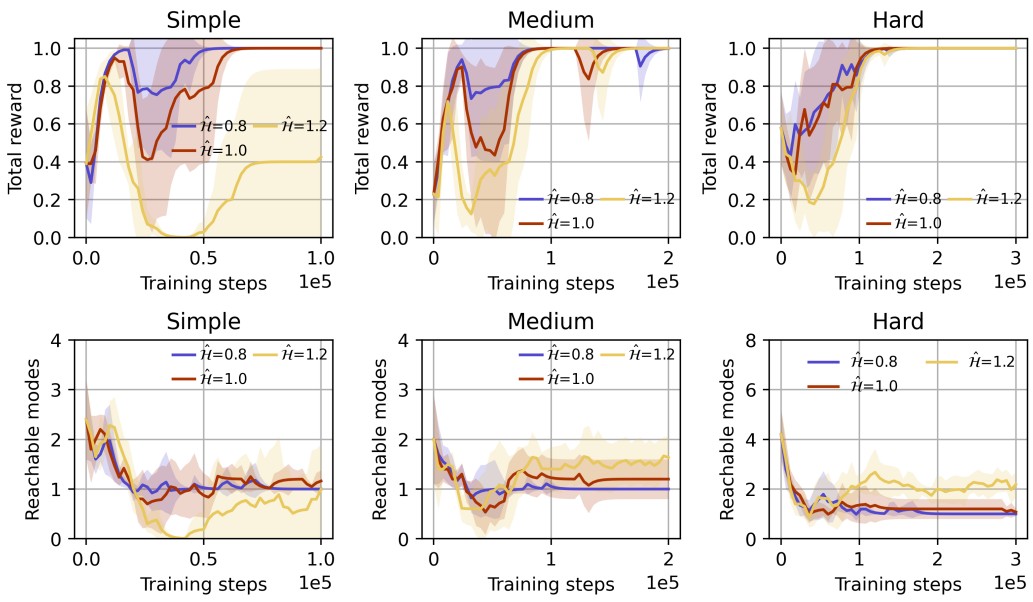

Figure 16: Learning curves of SAC in multi-goal PointMaze with varied temperatures.

Learning curves of S$^2$AC with varied temperatures

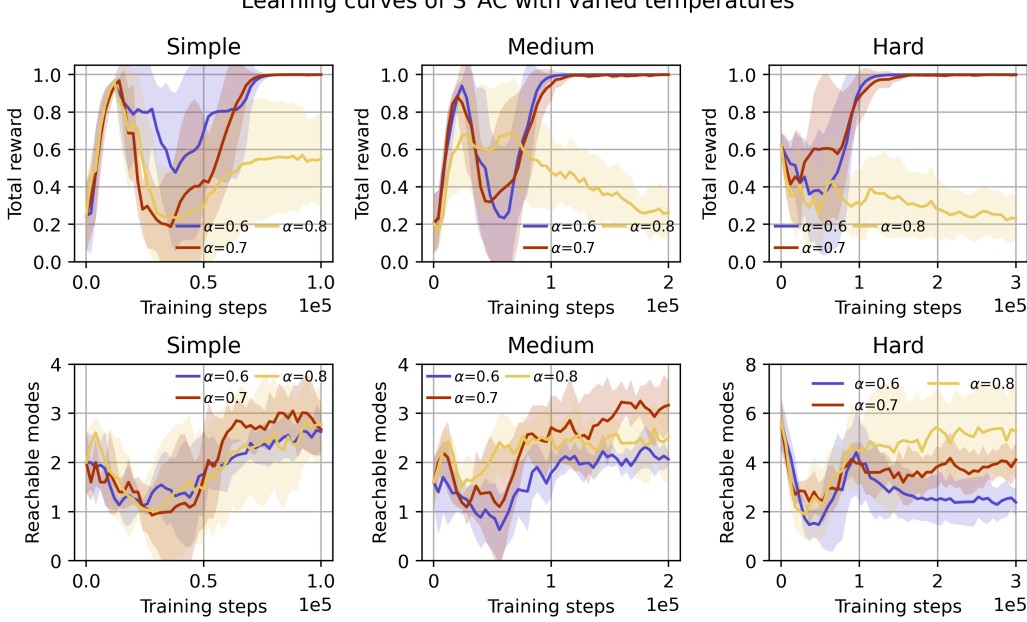

Figure 17: Learning curves of S$^2$AC in multi-goal PointMaze with varied temperatures.

## B.4 Sensitivity of Temperature in Game Content Generation

We train each algorithm with three groups of temperatures, namely high, mid, low, to investigate the sensitivity of temperature. The high temperature group uses the same temperature with the multi-goal PointMaze. The low temperature group are set to $-|\mathcal{A}|$ for SAC and DACER and near zero for DrAC, SQL and S$^2$AC. The mid temperature group use temperatures averaged between low and high groups. Fig. 18 shows the algorithm performance across temperatures.

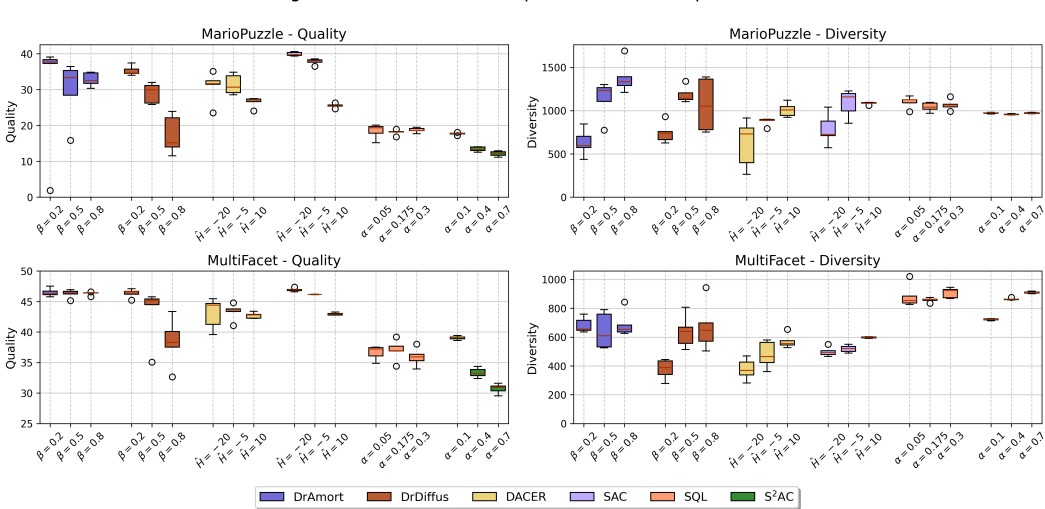

Figure 18: Sensitivity of temperature across tasks and metrics. The ticks of $x$-axis details the temperature.

Generally, the quality decreases along with raising temperature while diversity increases along with raising temperature. To provide an overall evaluation, we treat all policies trained by each algorithm across temperatures and seeds as a population, and compute the hypervolume (HV) metric for these populations [60]. The reference point is set according to the lowest quality and diversity performance across all algorithms, temperatures and seeds.

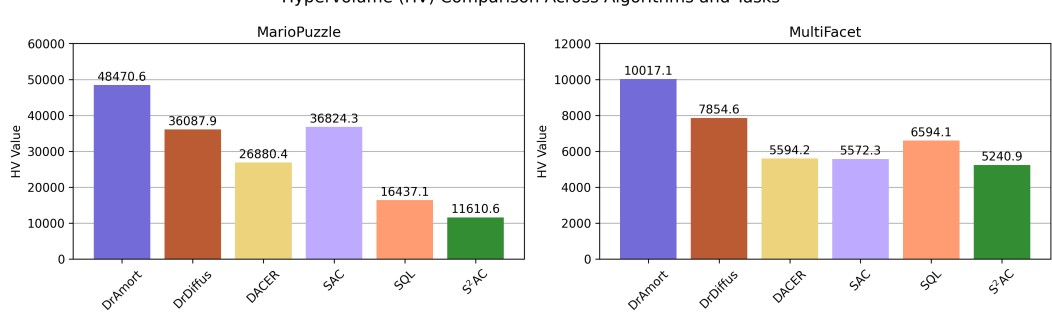

Figure 19: HV comparison of policy populations obtained by training each algorithm through three different temperatures.

The results show DrAmort produces the best population in terms of the HV metric, DrDiffus outperform baselines in MultiFacet and shows competitive performance in MarioPuzzle.

## B.5 t-SNE visualization of Generated Game Levels

To compare the diversity of levels generated by each algorithm more intuitively, we embed levels generated by each algorithm with t-SNE [46], and plot the embeddings in Fig. 20 and Fig. 21.

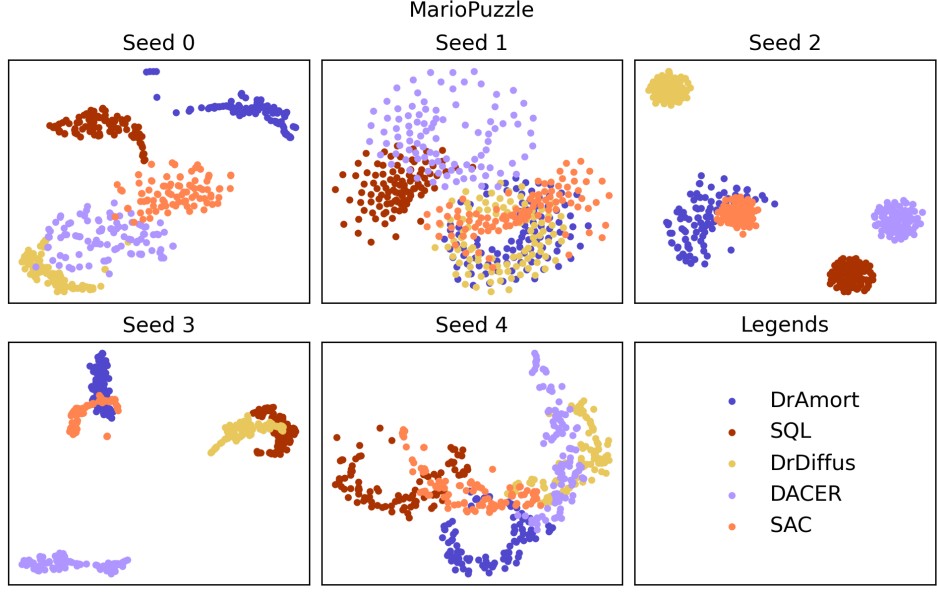

Figure 20: t-SNE embeddings of levels generated by each algorithm, under MarioPuzzle style.

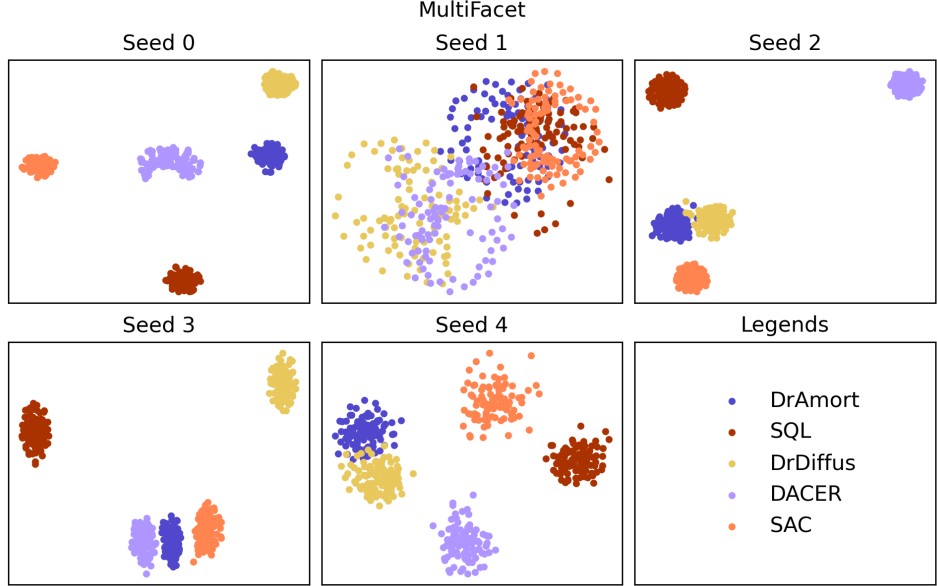

Figure 21: t-SNE embeddings of levels generated by each algorithm, under MultiFacet style.

Observing the embeddings, differences in terms of diversity are not significant. Different algorithms generally form different clusters. This may indicate that there is still ample room to improve the diversity with little harm to quality. Future work may investigate how to unleash the expressivity of multimodal policies more completely. Besides, average pairwise Hamming distance is a vanilla method to measure diversity of game content, more in-depth quantitative analysis may be considered in future work.

## B.6 Sensitivity of Temperature Hyperparameter in MuJoCo

We demonstrate the sensitivity of temperature hyperparameter of DrAmort and DrDiffus as follows.

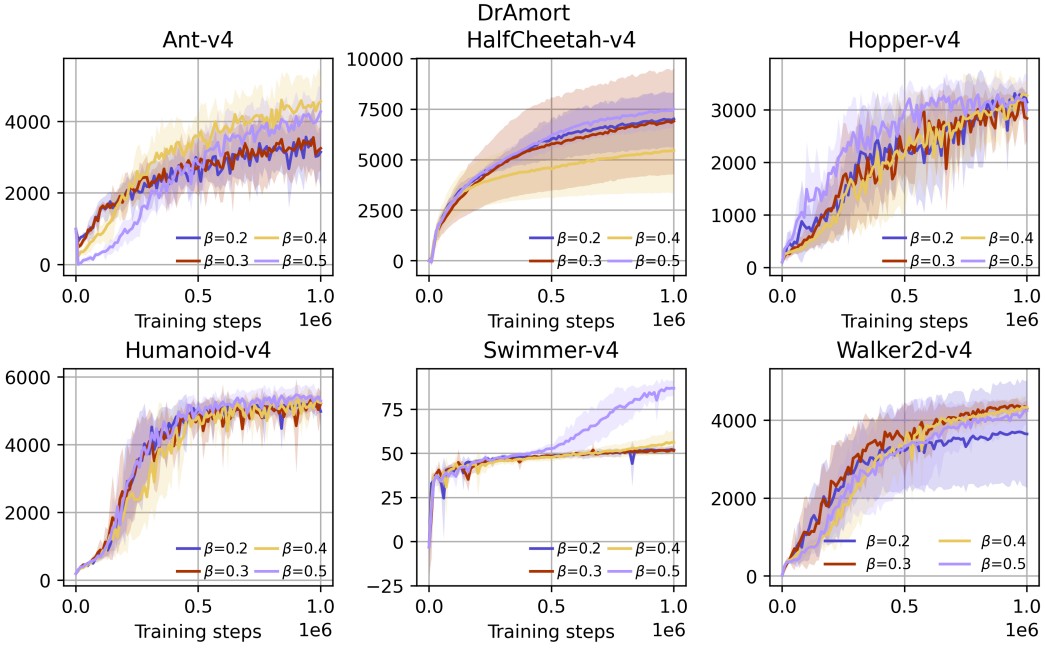

Figure 22: Learning curves of DrAmort with varied temperatures in MuJoCo

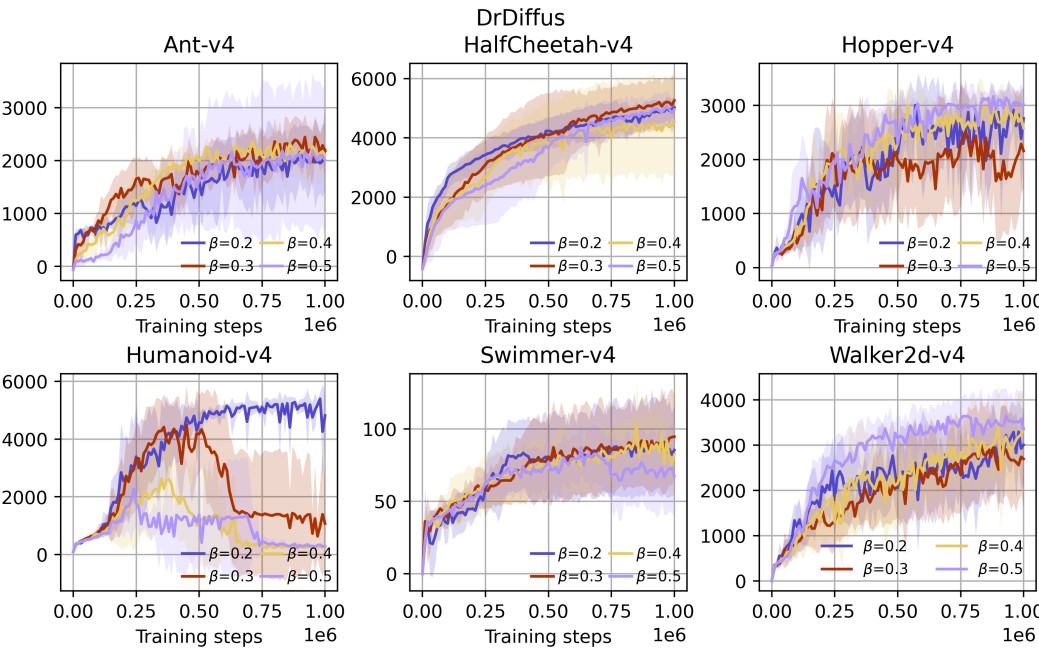

Figure 23: Learning curves of DrDiffus with varied temperatures in MuJoCo

DrAmort is robust to the temperature, overall $\beta = 0.5$ deliveries superior performance. DrDiffus suffers from a performance drop in Humanoid with $\beta > 0.2$. More investigation into this phenomenon is an important future work.

