# OpenReview forum: "Learning Intractable Multimodal Policies with Reparameterization and Diversity Regularization"
_NeurIPS.cc/2025/Conference — NeurIPS 2025 poster_

### Official Review · Reviewer_76hu · 2025-06-29

**Clarity:** 2
**Significance:** 1
**Originality:** 2
**Rating:** 3
**Confidence:** 4

**Summary:**

The paper proposes a policy gradient method for multimodal actors, and distance-based diversity regularization to boost exploration for diverse policies. Experiments are mainly conducted on two diversity-critical benchmarks, to show the improved diversity of the policies and learning performances. Experiments on conventional benchmark MuJoCo show the worse performances of the proposed methods against unimodal baseline SAC.

**Questions:**

In Fig. 5 (a), why are there a drop in performances in other baselines for simple and medium environments?

How much additional computation is introduced by distance-based diversity measure during training?

**Ethical Concerns:**

["NO or VERY MINOR ethics concerns only"]

**Final Justification:**

Thanks for the rebuttal. Given the unaddressed concerns regarding why amortized actor is better than diffusion actor, the lack of baseline like DPPO method, and unclear whether the diffusion gradients of policy value and entropy term are carefully handled, I keep my current score.

**Limitations:**

yes

**Quality:**

2

**Strengths And Weaknesses:**

Strengths:

The paper proposes a new sampling-based diversity measure for regularization purpose.

The two diversity-critical benchmarks are more proper to evaluate the multimodal policy representation.

The experiment analysis is quite comprehensive.

Weakness:

Fig. 2 is hard to interpret, I cannot understand how Fig. 2(a) unifies the two types of actors.

The unification of amortized and diffusion actors with reparameterized policy gradient seems trivial.

The paper mentions that the sampling-based diversity measure can be parallelized over GPUs for generating multiple samples, but this inevitably introduces additional computational costs during training.

On traditional MoJoCo benchmarks, the proposed methods perform worse than unimodal policy SAC.

Missing references and baselines. The multimodal policy representation in RL has been studied for a long time, and I think there are quite a lot of papers missing as references and baselines from this paper. Discussion and comparisons against these baselines would be necessary to prove the advantages of the proposed methods.

For mixture-of-Gaussian:

[1] Peng, X. B., Chang, M., Zhang, G., Abbeel, P., and Levine, S. MCP: learning composable hierarchical control with multiplicative compositional policies.

[2] Ren, Jie, et al. "Probabilistic mixture-of-experts for efficient deep reinforcement learning.

For gradient optimization on diffusion policy:

[3] Ren, Allen Z., et al. "Diffusion policy policy optimization.

[4] Chen, Yuhui, Haoran Li, and Dongbin Zhao. "Boosting continuous control with consistency policy."

[5] Ding, Zihan, and Chi Jin. "Consistency models as a rich and efficient policy class for reinforcement learning."

---

> ### Author Rebuttal · Authors · 2025-07-31
>
> We thank the reviewer for the insightful comments and suggestions. We run additional experiments to address reviewer’s questions and improve the thoroughness of our work. Due to the character limit, we only present most relevant and informative results using tables here. All presented results are averaged over 5 training seeds.
>
> **Regarding reviewer's questions:**
>
> **Q1:**
> Those baselines suffering from a performance drop are entropy-regularized algorithms. Based on the periodic visualization of trajectories in the mazes, we observed that the performance drop of those baselines is accompanied by exploration behaviours. At that stage, these algorithms appear to “wobble” among different goals. Possibly the gradient of entropy bonus pushes the actor to other directions and makes the policy explore in the maze and fails to reach goals. DACER handles entropy regularization by auto-adjusting the scale of an auxiliary noise instead of back-propagating the gradient of entropy regularization. Moreover, the computational cost of adjusting the scale is high, making such adjustment usually perform at a very low frequency (by default, DACER updates the scale every 10000 exploration steps). This possibly be the reason why DACER does not suffer from a performance drop. A more in-depth investigation shall be future work.
>
> **Q2:**
> Let $n$ ($n=8$ in our experiments) denote the number of diversity estimation pairs, we summarize the number of forward inferences and back-propagation per transition in one single update as follows.
>
> |  | Actor forward | Actor backward | Critic forward | Critic backward |
> | --- | --- | --- | --- | --- |
> | w/o diversity regularization | 2 | 1 | 3 | 2 |
> | w/ diversity regularization | 4n+2 | 2n+1 | 3 | 2 |
>
> The diversity estimation introduces $4n$ additional forward inference and $2n$ additional back-propagation, without additional forward and backward of the critic. As the neural networks used for continuous control are typically small (e.g., SAC uses MLPs with two 256-dimensional hidden layers by default), we empirically found DrAmort and DrDiffusion do not induce significantly more training times and memory usage, as displayed in the following table.
>
> >This table shows average time cost per 1000 training steps in Ant-v4 environment. The test is conducted on a Linux server with 8 NVIDIA RTX 3090 GPUs and an Intel Xeon Platinum 8375C CPU. For each algorithm, we run three trials on GPU cards 0, 1 and 2, respectively, and then average the training time per 1000 steps. All algorithms are implemented with PyTorch 2.5.1, and the CUDA version is 12.5.
> |  | SAC | SQL | DrAmort | DACER | DrDiffus | S2AC |
> | --- | --- | --- | --- | --- | --- | --- |
> | Time per 100 training steps | 22.7s | 18.9s | 21.0s | 76.9s | 84.8s | 220.5s |
> | Peak GPU memory usage | 352M | 472M | 392M | 364M | 608M | 3462M |
>
> According to the table, we found that DrAmort is even faster than SAC. The reason is that SAC requires computing and back-propagating the log-probability at certain actions, which requires more sequential tensor operations. Though DrAmort requires more FLOPs, the computation is parallelized and takes fewer sequential tensor operations. As a result, DrAmort uses even fewer GPU clock ticks and is slightly faster than SAC, given that the neural network is sufficiently small compared to the GPU’s parallel computation capability. For DrDiffus, it takes about 10% more time for training compared to DACER, but its number of reachable goals in multi-goal PointMaze is much higher.
>
> For large-scale tasks that require larger networks, such as visual control, we can inject the latent variable at the last few layers. For example, if a CNN is used, the latent variables can be injected after the convolutional layers. In this case, the computation time and memory usage will not significantly increase.
>
> **Regarding specific weakness:**
>
> > The unification of amortized and diffusion actors with reparameterized policy gradient seems trivial.
>
> We would like to justify our contribution regarding this point as follows:
> 1. Previous practices in training amortized actors are based on SVGD [a,b]. The effectiveness of the reparameterized policy gradient for training amortized actors is underexplored. We highlight that the reparameterized policy gradient is effective for training amortized actors.
> 2. Previous works did not formally discuss the theoretical connections between backpropagating $Q$ and policy gradient.
> 3. By formulating diffusion actors and amortized actors into a unified framework, we demonstrate the versatility of reparameterized policy gradient for training intractable multimodal actors. This formulation is general and extendable. It serve as a useful tool for understanding and devising multimodal RL models and algorithms.
>
> We have revised section 4.1 and the contribution statement 1 in the introduction to make the contribution of this part clearer and accurate.
>
> [a] Haarnoja, Tuomas, et al. "Reinforcement learning with deep energy-based policies." *International conference on machine learning*. PMLR, 2017.
>
> [b] Messaoud, Safa, et al. "S$^2$AC: Energy-Based Reinforcement Learning with Stein Soft Actor Critic." *The Twelfth International Conference on Learning Representations*.
>
> > The paper mentions that the sampling-based diversity measure can be parallelized over GPUs for generating multiple samples, but this inevitably introduces additional computational costs during training.
>
> Though the estimation of our diversity measure inevitably introduces additional computations during training. The actual training time is similar to the baselines as those operations are fully parallelized. For large-scale tasks, we can inject the latent variable at the last few layers to avoid much additional computation (See the response to Q2).
>
> > On traditional MoJoCo benchmarks, the proposed methods perform worse than unimodal policy SAC.
>
> Our core contribution is training mutlimodal actors with optimized diversity, which is verified with the other two domains. The main purpose of our experiment in MuJoCo benchmarks is to show our algorithm achieves competitive performance against SAC in conventional tasks. It is not so quite surprising that DrAC does not outperform SAC, as the only goal in MuJoCo is to move faster, making multimodality less important. In addition, while amortized actors are fast and possess strong multimodal expressiveness, there is a lack of an efficient and effective algorithm to train them prior to our work. SQL performs badly on MuJoCo and S2AC costs much more time and memory usage. Our algorithm fills this research gap.
>
> Meanwhile, as suggested by other reviewers, we are conducting comprehensive ablation study, including in MuJoCo benchmarks. We have finished the training of DrAmort with other $\beta$ values in Ant and HalfCheetah. The results suggest that the reason that our algorithm does not outperforms SAC is probably insufficient parameter tuning
>
> | $\beta$ | Ant | HalfCheetah |
> | --- | --- | --- |
> | 0.15 | 2513.8 | 5487.0 |
> | 0.25 | 3327.9 | 5640.9 |
> | 0.35 | 3440.2 | 6558.2 |
> | SAC | 3422.7 | 5515.7 |
>
> > Missing references and baselines. The multimodal policy representation in RL has been studied for a long time, and I think there are quite a lot of papers missing as references and baselines from this paper. Discussion and comparisons against these baselines would be necessary to prove the advantages of the proposed methods.
>
> We have revised our paper to discuss related works more comprehensively.
>
> To address reviewer’s concern, we have tested PMOE [2] and CPQL [4] in our multi-goal environment. For CPQL, we run the official code with default settings as it does not provide a hyperparameter to trade off between return and diversity. For PMOE, we perform grid search as we done for other algorithms, and picked the highest target entropy that ensures it achieves the optimal success rate. The results in the Medium maze are presented as follows.
>
> |  | Success Rate | Reachable Modes | Robustness-Removal | Robustness-Obstacle |
> | --- | --- | --- | --- | --- |
> | DrAmort | 1.00 | 3.2 | 0.98 | 0.48 |
> | DrDiffus | 1.00 | 1.6 | 0.65 | 0.10 |
> | PMOE | 1.00 | 2.0 | 0.82 | 0.15 |
> | CPQL | 1.00 | 1.0 | 0.50 | 0.00 |
>
> According to the results, CPQL does not exhibit diversity. PMOE demonstrates diversity but our DrAmort outperforms PMOE. Moreover, PMOE trains tractable mixture-of-Gaussian policies that only represent a fixed number of modes, while our algorithm is able to train intractable policies that can express arbitrary number of modes.

---

> > ### Comment · Reviewer_76hu · 2025-08-04
> >
> > I've read through the author's complete rebuttal, but still confused on why Amortized actor can perform better than Diffusion actor. And I doubt that the current observation is only feasible when the diversity of data distribution is not high enough (small scale environments), or when the diffusion policy gradient is not carefully treated. The reason for this is that the current practice in computer vision or several other domains, diffusion model is clearly more expressive than the additional condition on latent variable. For the sample based estimation of diversity, I'm not sure how it works for diffusion actor, as it takes multi-step $N$ for generating $x_0$, which indicates the gradient of such diversity term will also backpropagate the model at $N$-th order. The references I mentioned have careful treatment to avoid gradient vanishing/explosion in such case. With such careful treatment, the multimodal expressiveness of diffusion policy is generally quite good, and I didn't see a clear reason why this does not work well unless with technical flaws.
> >
> > I do not view the theoretical bridging of the amortized actor and diffusion actor as a significant contribution, and those previous work for optimizing diffusion policy (although without diversity bonus) should serve as a necessary baseline as they are naturally multimodal.

---

> > > ### Author Response · Authors · 2025-08-08
> > >
> > > Thank you for your comments. We would like to justify several points.
> > >
> > > (1) We understand that the phenomena that amortized actor can outperform diffusion actor might be counter-intuitive.  However, online continuous RL is a quite different domain, where amortized actor is less explored. To our best knowledge, there is no evidence showing that amortized actor should be worse than diffusion actor in online RL.
> > >
> > > (2) We do have larger scale environment, which is game content generation (Section 5.2).
> > >
> > > (3) Our diversity regularization will not involve higher-order back-propagation, as gradients are averaged over pairwise samples.
> > >
> > > (4) We have carefully adopted the diffusion actor implementation of DACER and we have made our comparison fair. DACER is a state-of-the-art online RL algorithm with diversity bonus. DACER is newer than [4] and [5]. As for [3], it focuses on offline-to-online setting. As our main contribution is a general learning framework for training intractable multimodal actors with diversity regularization, we would like to pose “why sometimes amortized actors outperform diffusion actors” as an open question for future research.
> > >
> > > (5) We proposed a general learning framework with a novelty diversity regularization, the theoretical bridging is only one of the ingredients of our method. We will rephrase related content to improve clarity.
> > >
> > > (6) Regarding baselines, we have tested CPQL [4] during the rebuttal, it shows little diversity and only reaches one goal in multi-goal maze environment. Without diversity bonus, the policy tends to be more and more greedy and nearly deterministic. As we focus on scenarios that requires diversity, we did not compare these algorithms without diversity bonus.

---

### Official Review · Reviewer_yEqZ · 2025-06-30

**Clarity:** 3
**Significance:** 2
**Originality:** 2
**Rating:** 4
**Confidence:** 4

**Summary:**

This work proposes an actor-critic RL algorithm, Diversity-regularized actor-critic (DrAC), to train multimodal policies. The authors motivate their approach by noting that while previous works have employed generative models such as diffusion models as policies, they are limited by the fact that computing policy likelihoods for such models is intractable. To solve this issue, DrAC considers a parameterization that maps current states and a latent noise vector to the action, which also includes existing works on diffusion models. The authors propose training these actors through PGRep, formalized in Proposition 1. Experiments are performed in a modified PointMaze environment to study multimodal behaviours, game content generation to study the effectiveness of DrAC for generative tasks, and MuJoCo benchmark for continuous control tasks.

**Questions:**

- It is somewhat surprising that DrAmort works better than DrDiffus, since one would expect diffusion models to be more powerful samplers compared to a simple neural network conditioned on a noisy latent. The authors offer one possible explanation, that the number of denoising steps is too low. Could the authors try with more diffusion steps to get a sense of how well DrDiffus (and the other diffusion methods) can perform if we disregard the extra compute?
- The parameter $\beta$ seems like a crucial component of DrAC since it controls the trade-off between diversity and optimizing the Q-function. Could the authors discuss how sensitive DrAC is to $\beta$ and how to tune this parameter in practice? Including an ablation study would also strengthen the empirical contribution.
- Another parameter that warrants a discussion is the number of diversity estimation pairs. How does increasing the number of pairs affect the performance of DrAC?
- From Table 2, the authors use the target entropy for SAC and DACER to be $0.5|\mathcal{A}|$, whereas the standard practice is to use $|\mathcal{A}|$. Could the authors explain this particular choice?
- Also from Table 2, the authors use a fixed entropy coefficient for DACER in the multi-goal and game generation experiments, whereas the paper uses automatic temperature tuning like SAC. Could the authors explain this particular choice?
- The authors mention (line 318) that DACER uses a more advanced critic learning technique. Could the authors elaborate on this? From their paper, it looks like they use standard double Q-learning.
- Line 220: the authors state that determining the range of $D_\theta$ is trivial, and they introduce $\beta$ to “ease the triviality”. Could the authors explain what they mean here?

**Minor errors and suggestions:**

- Line 132 should also include the citation for Ziebart et al. [X] when discussing maximum entropy RL.
- Line 134: the definition of $Q_\text{soft}$ should include entropy for all future states $\mathcal{H}(\cdot | s_{t+k})$
- Line 160: in Equation 6, the model $\epsilon_\theta$ does not predict the mean, it predicts the noise at each step.
- Line 189: “The geometric mean is sensitive to small values as production is taken instead of summation” is grammatically incorrect - “production” is not a valid word to refer to the product of several terms.

**Ethical Concerns:**

["NO or VERY MINOR ethics concerns only"]

**Final Justification:**

During the rebuttal period, the authors provided ablation studies on the two main hyperparameters and an additional comparison between the diffusion actor and the amortized actor. These results improved the empirical contribution of the work.

The main concerns were (1) regarding the definition of diversity adopted in the paper, and whether it is a useful proxy for entropy (as was motivated in the paper), and (2) Proposition 1 being a restatement of the deterministic policy gradient theorem.

During the discussions, the authors indicated that the goal was *not* to suggest a proxy for the entropy, and provided some empirical evidence that increasing the diversity using their method leads to an increase in entropy. The authors also assured that Proposition 1 will be rewritten as an equation since it is a known result. In light of these changes, I update my rating to weak accept.

**Limitations:**

The authors discuss avenues for future work, including improved architectures, different diversity metrics, and temperature schedules. However, there is no discussion on the limitations of their approach. Some points to include here could be the strong full-rank assumption in Proposition 1, the fact that the diversity surrogate may not align with the true entropy of the policy, and the dependence of DrAC on the parameter $\beta$.

**Paper Formatting Concerns:**

The paper seems to adhere to the NeurIPS formatting instructions.

**Quality:**

2

**Strengths And Weaknesses:**

**Strengths**

- This work unifies amortized actors and diffusion actors under a single framework, which could be useful for studying such parameterizations.
- The paper contains diverse experiments, demonstrating that DrAC elicits multimodal behaviours, generates diverse latent vectors for a generative model, and performs competitively with other methods on continuous control tasks.
- The authors have taken care to use the same network architecture across all methods for comparison. They also tune SAC properly on the MuJoCo benchmark, matching my personal observation that it is indeed a very strong baseline and most methods do not actually outperform SAC on these tasks.
- The authors have provided a detailed hyperparameter table for DrAC and baselines in the appendix, as well as provided the code in the supplementary material. While I did not study the code, the details in the appendix are quite thorough, and the authors’ efforts in promoting reproducibility are appreciated.

**Weaknesses**

- Proposition 1 asserts that any actor expressible as a deterministic mapping $a = f_\theta(s,z)$, where $z$ is sampled from some noise distribution, can be trained with the gradient $\nabla_a Q \nabla_\theta f_\theta$. This seems to be a restatement of the deterministic policy gradient theorem [1], except that the policy takes an additional input. As noted by the authors, existing methods like DACER (and also Diffusion QL [2] in the offline setting) use DPG to train actors. Because this practice of taking gradients through the Q-function is well understood and widely used, could the authors clarify what conceptual or algorithmic advance is provided by PGRep?
- The assumption in Proposition 1 that $\nabla_z f_\theta(s, z)$ is full rank for all $s \in \mathcal{S}$ is a very strong assumption. It is well known that the gradients of a neural network develop a low-rank structure [3], which means the conditions for Proposition 1 will almost never be satisfied in practice.
- The paper adopts a very specific operational definition of diversity and makes two claims that merit reconsideration.
    - Lines 46–50 and 106–108 state that DACER is *unique* in accounting for decision diversity because it injects Gaussian noise into the sampled action. This overlooks the fact that any actor that samples from a multimodal policy (e.g., diffusion actors trained to sample approximately from the Boltzmann distribution of $Q$ [4,5,6,7]) inherently produces action diversity without additional noise, and DACER’s unimodal Gaussian perturbation does not necessarily translate into richer behavioural diversity. The current wording, therefore, understates prior work that achieves diversity implicitly.
    - Diversity is measured via the geometric mean of pairwise action distances, which favours a single, broadly dispersed cloud over well-separated clusters. In many RL settings, separated clusters better capture multimodal behaviour, with each cluster corresponding to a distinct strategy, whereas a diffuse unimodal cloud may just reflect stochastic deviation around one mode. The paper would benefit from explaining why a “balanced spread” is preferable to explicit mode coverage. Since this regularizer is intended as a proxy for entropy, an empirical check would help: e.g., compute the actual entropy of the two distributions in Figure 3 - does distribution X indeed have higher entropy than distribution Y? Demonstrating that the geometric mean aligns more closely with entropy than alternative measures would strengthen the motivation for this choice.
- Some closely related works [6,7] also learn a policy that samples from the Boltzmann distribution of the Q-function, and are not discussed. In particular, DQS [6] also exhibits multimodal behavior in a 2D navigation setting with sparse rewards, very similar to the experiment in Section 5.1

*[1] Silver, David, Guy Lever, Nicolas Heess, Thomas Degris, Daan Wierstra, and Martin Riedmiller. "Deterministic policy gradient algorithms." In International conference on machine learning, pp. 387-395. Pmlr, 2014.*

*[2] Wang, Zhendong, Jonathan J. Hunt, and Mingyuan Zhou. "Diffusion Policies as an Expressive Policy Class for Offline Reinforcement Learning." In The Eleventh International Conference on Learning Representations, 2023.*

*[3] Baker, Bradley T., Barak A. Pearlmutter, Robyn Miller, Vince D. Calhoun, and Sergey M. Plis. "Low-rank learning by design: the role of network architecture and activation linearity in gradient rank collapse." arXiv preprint arXiv:2402.06751 (2024).*

*[4] Psenka, Michael, Alejandro Escontrela, Pieter Abbeel, and Yi Ma. "Learning a diffusion model policy from rewards via q-score matching." arXiv preprint arXiv:2312.11752 (2023).*

*[5] Yang, Long, Zhixiong Huang, Fenghao Lei, Yucun Zhong, Yiming Yang, Cong Fang, Shiting Wen, Binbin Zhou, and Zhouchen Lin. "Policy representation via diffusion probability model for reinforcement learning." arXiv preprint arXiv:2305.13122 (2023).*

*[6] Jain, Vineet, Tara Akhound-Sadegh, and Siamak Ravanbakhsh. "Sampling from Energy-based Policies using Diffusion." arXiv preprint arXiv:2410.01312 (2024).*

*[7] Ishfaq, Haque, Guangyuan Wang, Sami Nur Islam, and Doina Precup. "Langevin Soft Actor Critic: Efficient Exploration through Uncertainty-Driven Critic Learning." arXiv preprint arXiv:2501.17827 (2025).*

---

> ### Author Rebuttal · Authors · 2025-07-31
>
> We thank the reviewer for the insightful comments and suggestions.  We run additional experiments to address reviewer’s questions and improve the thoroughness of our work. Due to the character limit, we only present most relevant and informative results using tables here. All presented results are averaged over 5 training seeds.
>
> **Regarding reviewer's questions:**
>
> **Q1:** Results in the Medium maze are as follows
>
> **DrDiffus**
> | Diffusion steps | Success Rate (SR) | Reachable Modes (RM) | Robustness-Removal (RR) | Robustness-Obstacle (RO) |
> | --- | --- | --- | --- | --- |
> | 10 | 1.00 | 1.2 | 0.59 | 0.13 |
> | 20 | 1.00 | 1.6 | 0.65 | 0.10 |
> | 30 | 1.00 | 1.4 | 0.69 | 0.14 |
> | 40 | 1.00 | 2.6 | 0.81 | 0.11 |
>
> **DACER**
> | Diffusion steps | SR | RM | RR | RO |
> | --- | --- | --- | --- | --- |
> | 10 | 1.00 | 1.0 | 0.50  | 0.10 |
> | 20 | 1.00 | 1.0 | 0.50 | 0.09 |
> | 30 | 1.00 | 1.0 | 0.50  | 0.12 |
> | 40 | 1.00 | 1.0 | 0.50  | 0.12 |
>
> The results demonstrate that the diversity-related performance of DrDiffus is improved by raising the number of diffusion steps. While for DACER, the diversity-related performance is not improved. This is probably due to that DACER controls diversity by scaling an additional Gaussian noise instead of optimizing diversity through gradients w.r.t. the actor's parameters.
>
> **Q2:** We conduct additional experiments for this question, existing results includes:
>
> **Multi-goal Maze:**
>
> **DrAmort**
> | beta | SR | RM | RR | RO |
> | --- | --- | --- | --- | --- |
> | 0.6 | 1.00 | 1.0 | 0.50 | 0.04 |
> | 0.7 | 1.00 | 1.4 | 0.56 | 0.01 |
> | 0.8 | 1.00 | 4.6 | 0.91 | 0.62 |
> | 0.9 | 0.41 | 7.4 | 0.60 | 0.24 |
> | 1.0 | 0.36  | 7.4 | 0.55 | 0.23 |
>
> **DrDiffus**
> | beta | SR | RM | RR | RO |
> | --- | --- | --- | --- | --- |
> | 0.6 | 1.00 | 1.6 | 0.58 | 0.02 |
> | 0.7 | 1.00 | 2.0 | 0.55 | 0.03 |
> | 0.8 | 1.00 | 2.4 | 0.65 | 0.21 |
> | 0.9 | 0.90 | 6.8 | 0.83 | 0.65 |
> | 1.0 | 0.45 | 7.2 | 0.57 | 0.25 |
>
> **Game Content Generation:**
>
> **DrAmort**
> | beta | Content Quality | Content Diversity |
> | --- | --- | --- |
> | 0.2 | 30.9 | 631.3 |
> | 0.5 | 29.9 | 1137.3 |
> | 0.8 | 32.8 | 1384.9 |
>
> **DrDiffus**
> | beta | Content Quality | Content Diversity |
> | --- | --- | --- |
> | 0.2 | 35.3 | 744.3 |
> | 0.5 | 23.1 | 1061.3 |
> | 0.8 | 21.1 | 1009.3 |
>
> **MuJoCo**
>
> | beta | Ant | HalfCheetah |
> | --- | --- | --- |
> | 0.15 | 2513.8 | 5487.0 |
> | 0.25 | 3327.9 | 5640.9 |
> | 0.35 | 3440.2 | 6558.2 |
> | SAC | 3422.7 | 5515.7 |
>
> We will add comprehensive results and analysis to study this question in our paper.
>
> Regarding how to tune the temperature hyperparameter, generally when we need better diversity or exploration we shall raise $\beta$, and when we need more exploitation we shall reduce $\beta$. Our definition $\beta$ is normalized so that the scale of $\beta$ is comparable to the per-dimension pairwise distance between actions (See the response to Q7).  So one can roughly decide the range of $\beta$ and may be able to set $\beta$ according to the expected pairwise distance of actions. We will detail this in our paper.
>
> **Q3:**  In Medium Maze:
>
> **DrAmort**
> | Number of pairs | SR | RM | RR | RO |
> | --- | --- | --- | --- | --- |
> | 4 | 98.9% | 3.88 | 98.8% | 51.7% |
> | 8 | 99.5% | 3.20 | 97.9% | 49.6% |
> | 12 | 99.2% | 3.76 | 97.7% | 49.8% |
>
> **DrDiffus**
> | Number of pairs | SR | RM | RR | RO |
> | --- | --- | --- | --- | --- |
> | 4 | 84.3% | 1.52 | 64.1% | 14.4% |
> | 8 | 100.0% | 1.64 | 62.8% | 10.6% |
> | 12 | 100.0% | 1.52 | 64.9% | 13.1% |
>
> The success rate of DrDiffus with $n=4$ is lower than that with larger values of $n$. Expect for this, the performances of DrDiffus and DrAmort are not sensitive to $n$. To provides some intuition for tuning $n$, it is most likely that raising $n$ will reduce the variance in estimating diversity, but usually it not needs to be very large.
>
> **Q4:**
> For multi-goal PointMaze, we conducted a grid search for each algorithm before the formal evaluation. Based on the grid search, we picked the highest temperature that ensures each algorithm reaches an optimal success rate, to enable a fair and meaningful comparison. $0.5|\mathcal{A}|$ is the highest temperature value we found that still ensure them reach the optimal success rate, so we picked this value.
>
> For game content generation, we just follows the temperature settings in multi-goal PointMaze.
>
> For the MuJoCo experiements, we kept the default target entropy value as described in their papers. Our Table 3 lists the temperature/target entropy/entropy coefficients for all algorithms in MuJoCo.
>
> **Q5:**
> This is a typo and we are very sorry for that. We used temperature tuning technique like SAC following the paper of DACER. DACER does not use fixed $\alpha$ so there should be N/A in the corresponding cell in Table 2, and $\alpha=0.6$ is for S2AC. We sincerely thank the reviewer for the careful review and have revised this error.
>
> **Q6:**
> Under equation (11) of the DACER paper (the NeurIPS 2024 camera-ready version), it is stated “we employ the tricks in DSAC [11, 10] to mitigate the problem of Q-value overestimation.” And we carefully checked their official code and found that they used such a technique. That technique was introduced in [a], which is a distributional critic. We have revised our paper to make it more clear.
>
> [a] Duan, Jingliang, et al. "Distributional soft actor-critic with three refinements." *IEEE Transactions on Pattern Analysis and Machine Intelligence* (2025).
>
> **Q7:**
> The auto-tuning technique operates on $\alpha$ to match the actual pairwise distance-based diversity meaure $D(\pi)$ with a target $\hat D$, i.e., $D(\pi)=\mathbb{E}[\log\delta(a^x, a^y)] \approx \hat D$. The diversity regularization $\mathbb{E}[\log\delta(a^x, a^y)]$ takes logarithms and it is not so straightforward to determine the proper range of it. By introducing $\beta$, we have $\mathbb{E}[\log\delta(a^x, a^y)] \approx \log(\beta|\sqrt{\mathcal{A}}|) \Rightarrow \mathbb{E}[\delta(a^x,a^y)]/\sqrt{|\mathcal{A}|} \approx \beta$, so the proper range is easier to understand and one may determine $\beta$ according to the promising value of distances between actions.
>
> **Regarding specific weaknesses:**
> > Proposition 1 asserts that ... could the authors clarify what conceptual or algorithmic advance is provided by PGRep?
>
> 1. Amortized actors are lightweight policy models with strong multimodal expressiveness. However, algorithm for training amortized actors is underexplored in the field of multimodal RL, and to our best knowledge, no work has trained amortized actors reparameterized policy gradient. We highlight that the reparameterized policy gradient is effective for training amortized actors.
> 2. Though there have been some diffusion policy-based RL algorithms back-propagating the gradient of the Q-function to update the diffusion policy, the theoretical interpretation from the perspective of reparameterized policy gradient is overlooked.
> 3. We introduce a unified formulation for a range of intractable actors and highlight that PGRep is applicable to all these actors. This formulation serves as a useful tool for understanding and designing multimodal RL models and algorithms.
>
> > The assumption in Proposition 1 that is full rank ... almost never be satisfied in practice.
>
> Thanks for raising this point. We acknowledge that this type of assumption, which serves as a sufficient condition to establish the result, is strong. Yet, we find that it is not necessary for the practical algorithm to work well, which is also a common theory-practice gap in deep RL literature, where theoretical guarantees often rely on stronger assumptions while practical algorithms succeed despite some violations [b]. We would like to emphasize that this provides insights for the development of our algorithm in Section 4.2 even when the assumption is not strictly satisfied, where our extensive experiments show that DrAC works effectively across diverse environments and backbone components. This consistent performance gains suggest that the core mechanism in DrAC is robust to potential violations of theoretical assumptions.
>
> In the updated version, we will discuss the implications for Proposition 1 and explore connections to rank-deficient approximations that may explain why the method works despite assumption violations, and also further contextualize this assumption within the broader RL literature, which is a very interesting future direction for our work to explore smoother assumptions (e.g., regularity conditions for the gradients). Despite this theoretical point, our work remains significant, which aims to develop a unified framework for multimodal actors with a novel distance-based regularization approach that achieves consistent performance gains in diversity-critical domains with improved generalization ability.
>
> [b] Tang, Y., & Agrawal, S. (2018). Boosting trust region policy optimization by normalizing flows policy. *arXiv preprint arXiv:1809.10326*.
>
> > Diversity is measured via the geometric mean of pairwise action distances, ... would strengthen the motivation for this choice.
>
> In Figure 2, the ground truth of $X$ is a Gaussian distribution with $\sigma=1$, while the ground truth of $Y$ is a mixture-of-Gaussian with uniform weights and two Gaussian components with $\sigma_1=\sigma_2=0.2$. We calculated their entropy: $\mathcal{H}(X)=2.838$,  and $\mathcal{H}(Y)\approx 0.3121$. This demonstrates that geometric mean aligns more closely with entropy compared with the vanilla average pairwise distance measure. We have updated Figure 3 to include such information.
>
> We agree that separated clusters better capture multimodal behaviour, while there still exists a trade-off, since overly small clusters might harm local exploration. The regularizer presented in Equation (8) is our best practice so far.
>
> **For other concerns about missed references, understatement for prior works, minor errors:** We thank for reviewer's time and rigorous review and will carefully revise them in our paper.

---

> > ### Comment · Reviewer_yEqZ · 2025-08-06
> >
> > Thank you for providing ablation experiments for $\beta$ and the number of diversity pairs, and clarifying implementation details of baselines. The ablation studies improve the empirical results in the paper.
> >
> > > Amortized model vs diffusion model performance
> >
> > I am still unsure why the single step amortized model works better than diffusion models, since this goes against the vast literature on generative models. This point is also noted by reviewer 76hu. It might be the experimental setting as noted by reviewer 76hu, or perhaps implementation details (like architecture, hyperparameters).
> >
> > > Significance of theoretical results
> >
> > I maintain that Proposition 1 is essentially the deterministic policy gradient theorem, and the policy taking an additional input (the noise latent variable) does not change the underlying derivation or significance. Though I do see the authors' point that the full rank assumption may not matter empirically, as this theory-experiment gap is common in deep learning.
> >
> > > Notion of diversity
> >
> > Thank you for providing the entropies of the two distributions in Figure 2. My comment was more general and not specific to this example - it is not obvious why the geometric mean is a good proxy for entropy (ideally, all experiments should include entropy values to confirm this method leads to higher entropy policies) and seems to lack a theoretical basis. As mentioned in my review, the statement about DACER being the *only* method that encourages diversity via additive Gaussian noise is misleading - any diffusion policy that learns a multimodal distribution can sample diverse actions.
> >
> > The authors might want to (i) convincingly demonstrate that their notion of diversity is preferable in the general case (not only for the example in Figure 2), (ii) rephrase or remove entirely Proposition 1 since I do not believe it is a new contribution, (iii) address why diffusion actors seem to be worse than the amortized actor since this goes against the vast amount of literature in generative modeling.

---

> > > ### Author Response · Authors · 2025-08-08
> > >
> > > Thank you for your follow-up, we would like to discuss these points one-by-one.
> > >
> > > (1) Our notion of diversity is not only a proxy of entropy. As previously discussed, sometimes multiple clusters would be preferred in multimodal RL. Given the definition of entropy  $\mathcal{H}=\int_x \log P(x) \mathrm{d}x$, it's definite that entropy does not take the distances between non-overlapped clusters into consideration at all. Our metric is based on distance so it might make policies easier to form multiple clusters/modes. We believe that entropy is not always the best diversity measure, and there is still much to explore with the diversity measure. Meanwhile, diffusion policy and amortized actor naturally represent multimodal policy, but without diversification technique, the policies tends to become more and more greedy and lose diversity, which can also harm the performance in conventional online RL setting.
> > >
> > > On the other hand, we observed that with higher temperature, our algorithm learns policies with higher entropy, demonstrating that our diversity measure is correlated to entropy. Specifically, we use the entropy estimation technique employed in DACER to estimate the entropy of policies leaned by DrAmort under different temperatures. This technique fits Gaussian mixture-models on learned action distributions, and then compute the entropy of the fitted Gaussian-mixture-model to estimate the entropy of action distributions. Results are presented as follows
> > >
> > > In MuJoCo tasks
> > >
> > > | $\beta$ | Ant | HalfCheetah |
> > > | --- | --- | --- |
> > > | 0.15 | -12.569 | -22.893 |
> > > | 0.25 | -8.715 | -19.644 |
> > > | 0.35 | -6.103 | -18.567 |
> > >
> > > In multi-goal Maze
> > >
> > > | $\beta$ | simple | medium | hard |
> > > | --- | --- | --- | --- |
> > > | 0.6 | 0.237 | 0.199 | 0.021 |
> > > | 0.7 | 0.656 | 0.617 | 0.497 |
> > > | 0.8 | 0.901 | 0.906 | 0.826 |
> > >
> > > The average entropy of our learned policy increases along with temperature (target diversity) increases.
> > >
> > > (2) We will rephrase Proposition 1, make it a single equation. Reparameterized policy gradient is only one of the ingredients of our method. We will focus on highlighting the versatility and effectiveness of reparameterized policy gradient for learning intractable multimodal actors (especially amortized actors), which has been overlooked in the community.
> > >
> > > (3) We understand that it might be counter-intuitive that diffusion actors seem to be worse than the amortized actor, as diffusion models show advantages against generative adversarial networks in generative modeling, and diffusion actors show promising performance in offline RL. However, online continuous RL is a different domain, where amortized actor is rarely explored. To our best knowledge, there is no evidence showing that amortized actor should be worse than diffusion actor in online RL. Our implementation of diffusion actor is based on DACER, and we have made our comparison fair. It is also possible that diffusion actors will exhibit advantages with deeper networks or better fine-tuning. However, as our main contribution is a general learning framework for training intractable multimodal actors with diversity regularization, we would like to pose “why diffusion actors is sometimes worse than the amortized actor” as an open question for future research.

---

> > > > ### Comment · Reviewer_yEqZ · 2025-08-08
> > > >
> > > > Thank you for the response and for clarifying that the goal was *not* to suggest a proxy for entropy but to propose a different diversity metric. Although this seems to differ from the phrasing in the paper, where the motivation appears to be that entropy can be challenging to estimate, hence this metric is used to overcome this challenge (lines 40, 180). I suggest the authors rephrase these parts and make this clear in the motivation.
> > > >
> > > > The results comparing the effect of temperature on the entropy are helpful and provide evidence that this parameter can control some notion of diversity in the underlying policy.
> > > >
> > > > The point about diffusion vs amortized actors is still somewhat confusing. However, based on the discussions, I will increase my rating provided that the changes above are incorporated and Proposition 1 is rewritten as an equation, as indicated by the authors in their response.

---

> > > > > ### Author Response · Authors · 2025-08-08
> > > > >
> > > > > Thank you for your constructive suggestions! We will carefully revise our paper, especially rephrase motivations and rewrite Proposition 1 as an equation. We believe addressing these points will significantly improve the quality and clarity of our paper. We will also add full hyperparameter sensitivity analysis and diversity analysis discussed during rebuttal, to improve the understanding of our empirical results.

---

### Official Review · Reviewer_C7Lq · 2025-06-30

**Clarity:** 2
**Significance:** 3
**Originality:** 2
**Rating:** 4
**Confidence:** 4

**Summary:**

This paper introduces PGRep, a reparameterized policy gradient method that works with a wide range of policies, including amortized and diffusion actors. It avoids using log-probabilities by adding a simple distance-based diversity regularizer, making it easier to train complex policies. The ideas build on existing methods like SAC and SQL, but the paper puts them together into a clear and useful framework. The experiments cover navigation, level generation, and continuous control tasks, showing strong results across settings.

**Questions:**

- For Figure 7(b), would you be able to provide **quality-diversity curves** for each algorithm by sweeping the diversity-regularisation coefficient (e.g., $\alpha$ or $\beta$) and plotting the resulting final return versus diversity?  This would reveal (i) how each method moves along the return–diversity frontier, and (ii) whether DrAC’s Pareto front consistently dominates those of the baselines.
- Related to the previous point, how sensitive is DrAC’s performance to the choice of $\alpha$/$\beta$?
- Figure 8 reports only episodic return on MuJoCo tasks.  To support the claim that DrAC delivers *both* quality and diversity, could you include an explicit diversity measure or visualization in this setting as well (e.g., action entropy, pairwise action distance, trajectory-level variation, or heat map)?

**Ethical Concerns:**

["NO or VERY MINOR ethics concerns only"]

**Final Justification:**

After reading the rebuttal, I still believe this is a technically solid paper and lean toward accepting it. Therefore, I've adjusted my confidence score accordingly.

**Limitations:**

yes

**Paper Formatting Concerns:**

I have reviewed the paper's formatting and found no issues.

**Quality:**

3

**Strengths And Weaknesses:**

**Strength**
- The paper provides a clear, general gradient formula that works for any stochastic-mapping actor, such as amortized or diffusion policies. This makes the method broadly applicable.
- The method works well on a range of tasks, including navigation, level generation, and continuous control. It often outperforms strong baselines, showing that the approach is robust.

**Weaknesses**
- The reparameterization trick is standard in SAC and SQL-style methods. While some diffusion-based approaches use similar ideas [1] for trajectory optimization, typical diffusion policies for imitation learning do not involve policy gradients or reparameterization.
- The benchmark for diversity measure is narrow. It only uses the reachable goal count as the diversity metric in evaluation. However, I think some diversity measures in conventional benchmarks are also important and insightful (see questions below).

[1] Huang, Zhiao, et al. "Reparameterized policy learning for multimodal trajectory optimization." International Conference on Machine Learning. PMLR, 2023.

**Typos**
- **Line 77** There is an extra period “.” at the end of the sentence—please remove it.
- **Line 142 (Eq. 3)** The first term inside the argmax should read
  $
  \alpha\,\varrho\bigl(\pi(\cdot|s)\bigr)
  $
  instead of
  $
  \alpha\,\varrho\bigl(\pi(\cdot|S_{t+k})\bigr).
  $
- **Line 197** Replace “i.d.d.” with the standard abbreviation “i.i.d.”

---

> ### Author Rebuttal · Authors · 2025-07-31
>
> We thank the reviewer for the insightful comments and suggestions. We run additional experiments to address reviewer’s questions and improve the thoroughness of our work. Due to the character limit, we only present most relevant and informative results using tables here. All presented results are averaged over 5 training seeds.
>
> **Regarding reviewer's questions:**:
>
> **Q1**: We are running all algorithms with different temperatures in the game content generation environment. We will add these figures in our revision. We have finished partial experiments, in which we train all algorithms with three temperatures (five seeds each) and test the quality and diversity of content generated by the trained policies. To provide informative response, we compute hyper volume (HV) metric [a] to evaluate the policy population obtained by each algorithm. Specifically, we treat the 15 policies learned by each algorithm with 3 temperatures and 5 seeds each as one population.We take the global lowest quality and global lowest diversity among all policies learned by all algorithm as the reference point, then compute the HV metric for each population. The result is displayed as follows. (Training of S$^2$AC has not finished as it is too computational intensive)
>
> **MarioPuzzle**
>
> | DrAmort | DrDiffus | SAC | DACER | SQL |
> | --- | --- | --- | --- | --- |
> | 48470.6 | 35318.0 | 36824.3 | 26880.4 | 17101.0 |
>
> According to the table, our DrAmort demonstrates highest HV value. The HV values of DrDiffus and SAC are closed.
>
> **MultiFacet**
>
> | DrAmort | DrDiffus | SAC | DACER | SQL |
> | --- | --- | --- | --- | --- |
> | 7559.7 | 6402.5 | 4157.8 | 3957.7 | 3603.0 |
>
> According to the table, our DrAmort and DrDiffus achieve the top 2.
>
> Comprehensive experiment results will be discussed in our revised version, where we will also show visualization of Pareto fronts. Future work in generative RL for game content generation includes multi-objecive RL, where our method could serve as a promising base algorithm.
>
> [a] Zitzler, Eckart, and Lothar Thiele. "Multiobjective evolutionary algorithms: a comparative case study and the strength Pareto approach." *IEEE transactions on Evolutionary Computation* 3.4 (2002): 257-271.
>
> **Q2**: We conduct additional experiments to investigate this question. Based finished experiments, we present some results below to respond to this question. We will add comprehensive results and analysis in our paper.
>
> **Multi-goal Maze**
>
> We run DrAC with different values of $\beta$ in the our multipoint-goal maze environments, the final performances in the Hard maze are presented as follows.
>
> **DrAmort**
>
> | beta | Success Rate | Reachable Modes | Robustness-Removel | Robustness-Obstacle |
> | --- | --- | --- | --- | --- |
> | 0.6 | 1.00 | 1.0 | 0.50 | 0.04 |
> | 0.7 | 1.00 | 1.4 | 0.56 | 0.01 |
> | 0.8 | 1.00 | 4.6 | 0.91 | 0.62 |
> | 0.9 | 0.41 | 7.4 | 0.60 | 0.24 |
> | 1.0 | 0.36 | 7.4 | 0.55 | 0.23 |
>
> **DrDiffus**
>
> | beta | Success Rate | Reachable Modes | Robustness-Removel | Robustness-Obstacle |
> | --- | --- | --- | --- | --- |
> | 0.6 | 1.00 | 1.6 | 0.58 | 0.02 |
> | 0.7 | 1.00 | 2.0 | 0.55 | 0.03 |
> | 0.8 | 1.00 | 2.4 | 0.65 | 0.21 |
> | 0.9 | 0.90 | 6.8 | 0.83 | 0.65 |
> | 1.0 | 0.45 | 7.2 | 0.57 | 0.25 |
>
> According to the results, success rate is not sensitive to beta within a wide range  (about <0.8 for DrAmort and <0.9 for DrDiffus). diversity-related performance is sensitive to beta. Higher $\beta$ values enable DrAC to reach more modes and enhance few-shot robustness. However, overly high temperatures reduce the success rate, which also harms robustness metrics. We will discuss the sensitivity of beta comprehensively in our revised version.
>
> **Game content generation**: We finished evaluation with three $\beta$ values for MarioPuzzle style. Results are presented below
>
> **DrAmort**
> | beta | Content Quality | Content Diversity |
> | --- | --- | --- |
> | 0.2 | 30.9 | 631.3 |
> | 0.5 | 29.9 | 1137.3 |
> | 0.8 | 32.8 | 1384.9 |
>
> **DrDiffus**
> | beta | Content Quality | Content Diversity |
> | --- | --- | --- |
> | 0.2 | 35.3 | 744.3 |
> | 0.5 | 23.1 | 1061.3 |
> | 0.8 | 21.1 | 1009.3 |
>
> The quality of DrAmort is not sensitive to $\beta$, while diversity increases by raising $\beta$. For DrDiffus, quality increases and diversity decreases from $\beta=0.2$ to $\beta=0.5$, but $\beta=0.8$ does not show significant difference with $\beta=0.5$
>
> There are two aspects may be related to the experiment phenomenon: (1) higher temperature also improves exploration which could be helpful to quality (2) policy diversity is correlated but not identical to the diversity of generated levels. More fine-grained ablation study will be conducted and discussed in our revision. Future work in generative RL for game content generation includes multi-objecive RL, where our method could serve as a promising base algorithm.
>
> **MuJoCo**
>
> We are running additional experiments to analyze the sensitivity of DrAC in MuJoCo. Such experiments will be added to our revised version. We have finished partial experiments now. The performances of DrAmort with different values of $\beta$, in the first two MuJoCo environments, are presented as follows
>
> | beta | Ant | HalfCheetah |
> | --- | --- | --- |
> | 0.15 | 2513.8 | 5487.0 |
> | 0.25 | 3327.9 | 5640.9 |
> | 0.35 | 3440.2 | 6558.2 |
> | SAC | 3422.7 | 5515.7 |
>
> Performance of SAC is included as a baseline. According to the table, $\beta$ values that are higher than our original setting ($\beta=0.15$) improve the performance in Ant and HalfCheetah. Notably, the improvement in Ant from $\beta=0.15$ to $\beta=0.35$ is >36%, and with $\beta=0.35$, DrAmort outperforms SAC by >18% in HalfCheetah. More elaborate self-adaptation/annealing techniques could be researched in future work.
>
> **Q3**:
> We train DrAmort and DrDiffus with a higher $\beta$ in Ant, and records the trajectories by evaluating the final policies. We compute average distance among those trajectories as a diversity measure, as presented below.
>
> | $\beta$ | DrAmort | DrDiffus |
> | --- | --- | --- |
> | 0.15 | 3736.6 | 7962.8 |
> | 0.35 | 5463.3 | 8676.6 |
>
> According to the table, higher $\beta$ make trajectory diversity increase, though the only objective in MuJoCo is moving forward faster. We will visualize the trajectories in our revision.
>
> **Regarding specific weakness:**
> > The reparameterization trick is standard in SAC and SQL-style methods. While some diffusion-based approaches use similar ideas [1] for trajectory optimization, typical diffusion policies for imitation learning do not involve policy gradients or reparameterization.
> >
> > [1] Huang, Zhiao, et al. "Reparameterized policy learning for multimodal trajectory optimization." International Conference on Machine Learning. PMLR, 2023.
>
> We  have revised our paper to discuss [1]. We would like to clarify the novelty and significance here as follows:
>
> 1. SAC applies reparameterization trick but only for tractable (squashed) Gaussian actors. SQL employs Stein variational gradient descent to back-propagate the estimated gradient of the KL-divergence objective in entropy-regularized RL. This is different from our gradient estimator formulated in Equation (7). To our best knowledge, no work has trained amortized actors with the reparameterized policy gradient as formulated in our Equation (7) before. Our work highlights the overlooked effectiveness of training amortized actors through the reparameterized policy gradient. We also demonstrates that amortized actor is highly potential for learning multimodal policy, which is underexplored before.
> 2. Both [1] and our work adopt the reparameterization trick, however, [1] applies the reparameterization trick to estimate the gradient of an evidence lower bound, while our work applies the reparameterization trick to estimate the standard policy gradient. Our work  primarily differ from [1] in two aspects. (1) The algorithm proposed in [1] is a model-based algorithm that additionally learns a latent dynamics model, while our work focuses on the model-free setting; (2) Paper [1] reformulates the learning objective into an evidence lower bound of optimality, while our learning objective is expected diversity regularized return.
> 3. Recently, some online RL algorithms for diffusion policies have been proposed, in which [b,c] also backpropagates the gradient of $Q$. These methods show promising performances. However, the theoretical interpretation of this operation from the perspective of reparameterized policy gradient is overlooked. We highlights that backpropagating $Q$ is an estimator of policy gradient with reparameterization trick, which is applicable to all stochastic-mapping actors including diffusion actors and amortized actors. This formulation serves as a useful tool for understanding and designing online model-free multimodal RL algorithms.
>
> We have revised our paper to discuss related works more comprehensively and improve the clarity of our contributions
>
> [b] Wang, Yinuo, et al. "Diffusion actor-critic with entropy regulator." *Advances in Neural Information Processing Systems* 37 (2024): 54183-54204.
>
> [c] Ding, Zihan, and Chi Jin. "Consistency Models as a Rich and Efficient Policy Class for Reinforcement Learning." *The Twelfth International Conference on Learning Representations*.
>
> **For typos:**  We thank for reviewer's time and rigorous review. We will carefully revise them and further proofread our paper rigorously.

---

> > ### Comment · Reviewer_C7Lq · 2025-08-02
> >
> > Thank you for the detailed rebuttal. After reading it, I still believe this is a technically solid paper and lean toward accepting it. Therefore, I've adjusted my confidence score accordingly.

---

> > > ### Author Response · Authors · 2025-08-03
> > >
> > > Thank you for your supportive comments and thank you for your constructive suggestions again. We will address the points raised during rebuttal carefully and holistically in revision.

---

### Official Review · Reviewer_T7pd · 2025-07-02

**Clarity:** 3
**Significance:** 3
**Originality:** 2
**Rating:** 5
**Confidence:** 4

**Summary:**

This paper proposes a class of diversity-regularized actor-critic (DrAC) algorithms for learning multimodal policies in RL with continuous actions. DrAC assumes a reparameterizable policy class (including amortized and diffusion actors/policies) and uses the reparameterization trick for calculating the policy gradient. The diversity of actions given a state is promoted by a geometric-based distance loss for a pair of generated actions. The policy optimization loss and this distance loss are balanced by a coefficient that is tuned in a similar way to SAC’s temperature. Two instances of DrAC algorithms based on amortized and diffusion actors, respectively, are empirically evaluated against some common baselines. In multi-goal environments and generative tasks, the amortized variant is shown to be generally better, achieving both high return and more diverse behaviors, while the diffusion variant is shown to be comparable to a diffusion baseline. In a conventional benchmark (MuJoCo), both variants and other amortized/diffusion-based baselines do not outperform SAC and are sometimes worse.

**Questions:**

Questions that may impact the evaluation:
1. For the multi-goal achieving experiment, could you elaborate on the process of tuning and selecting the temperature hyperparameter? What hyperparameters are tuned for each method?
2. How is $\beta$ tuned in the MuJoCo experiment?
3. Are DrAmort and DrDiffus sensitive to $\beta$ in different settings, respectively?

Questions or suggestions that do not impact the rating:
1. What is used as the ground truth density for Figure 3?
2. If I understand correctly, the actor in S$^2$AC should also be considered an amortized actor. However, I’m not sure if the current definition is broad enough to include this instance. Could you shed some light on this? If possible, extending the current definition to be more general would be helpful.
3. In Line 132, the introduction of the entropy-regularized RL framework is attributed to Haarnoja et al. (2018). However, at least Toussaint (2009) introduced such a framework under the name of probabilistic inference. The discussion around Line 132 should be updated to assign credit properly.

Toussaint, M. (2009, June). Robot trajectory optimization using approximate inference. In Proceedings of the 26th annual international conference on machine learning (pp. 1049-1056).

**Ethical Concerns:**

["NO or VERY MINOR ethics concerns only"]

**Final Justification:**

My main concerns around hyperparameter tuning and sensitivity to temperature are resolved. I believe with the new results and analysis promised in the rebuttal, the paper has become stronger.

I’ve also reviewed other reviews and the corresponding responses. While I agree that insufficient explanation of why amortized actors are better than diffusion actors is a weakness of the paper, I think the observation itself is already interesting.

Overall, I believe that this paper provides interesting results and contributions to the community, and I support the acceptance of the paper.

**Limitations:**

Yes.

**Paper Formatting Concerns:**

No.

**Quality:**

2

**Strengths And Weaknesses:**

The strengths of the paper include the following:
1. The paper provides some valuable insights about the utilities of more expressive policies in RL with continuous actions. On the one hand, unlike recent works that mostly focus on a simple 2D multi-goal toy problem [11, 28, 41], the paper demonstrates the utility of multimodal policies in two settings that are understudied in relevant work. On the other hand, it is surprising that, despite consistent claims of improvement for more expressive policies, they do not outperform the standard Gaussian policy in MuJoCo under the same neural network (NN) structure and hyperparameters. Both sets of results are interesting to many RL researchers who are curious about more expressive policies and call for further investigation on when they are useful.
2. The paper proposes a novel loss to encourage diversity for multimodal policies and demonstrates its effectiveness. The automatic coefficient adjustment also reduces the need to tune the relevant hyperparameters.
3. The paper is well-written, clear, and very easy to follow.

The paper also has some weaknesses:
1. The current results are suggestive and not thorough.
    - In the MuJoCo experiments, multimodal policies, being evaluated under the same neural network structure and hyperparameters as standard Gaussian policies, didn't outperform them. It's unclear whether this is due to insufficient hyperparameter tuning or if multimodal policies simply aren't necessary in these environments.
    - The process of tuning the temperature hyperparameters in the multi-goal experiment is not elaborated, potentially hindering the validity and rigorousness of the results.
2. There is no analysis on the sensitivity of the temperature hyperparameter $\beta$. There appear to be two values used in different settings (0.8 and 0.15), and little is known about their sensitivity. Further analysis on this could improve the understanding and applicability of the work.
3. One of the claimed contributions, specifically the derivation of policy gradient for reparameterizable policies, appears to be highly incremental to prior work (for example, Theorem 3 in Lan et al., 2022) and does not introduce substantial novelty beyond that. A more thorough review and discussion of existing work is necessary to accurately establish its place within the field. Considering other contributions, this is minor but still a weak point of the paper.

Lan, Q., Tosatto, S., Farrahi, H., & Mahmood, R. (2022). Model-free Policy Learning with Reward Gradients. In International Conference on Artificial Intelligence and Statistics (pp. 4217-4234). PMLR.

---

> ### Author Rebuttal · Authors · 2025-07-31
>
> We thank the reviewer for the insightful comments and suggestions. We run additional experiments to address reviewer’s questions and improve the thoroughness of our work. Due to the character limit, we only present most relevant and informative results using tables here. All presented results are averaged over 5 training seeds.
>
> **Regarding reviewer's questions:**
>
> **Q1**: We conducted a grid search for each algorithm before the formal evaluation. Based on the grid search, we picked the highest temperature that ensures each algorithm reaches an optimal success rate, to enable a fair and meaningful comparison. We have revised the paper to clarify this.
>
> **Q2**: As SAC is a standard baseline in MuJoCo, we compute the average log-distance (our diversity regularization metric) of the trained SAC agents with default target entropy $-|\mathcal{A}|$ in each MuJoCo task. Then, as we set target diversity for DrAC as $\hat D = \log(\beta \sqrt{|\mathcal{A}|})$, so we have $\beta \approx \exp(D(\pi))/\sqrt{|\mathcal{A}|}$, where $D(\pi)$ is the expected pairwse log-distance.  So we computed the corresponding $\beta$ value in each task as follows
>
> | Ant | HalfCheetah | Hopper | Humanoid | Swimmer | Walker2d |
> | --- | --- | --- | --- | --- | --- |
> | 0.1237 | 0.2133 | 0.1465 | 0.3555 | 0.1143 | 0.1784 |
>
> Except for Humanoid, all the results are around 0.15, so we chose $\beta=0.15$ to make the target diversity comparable with the default target entropy of SAC. We will also add an evaluation of the sensitivity to temperature in MuJoCo during the revision.
>
> **Q3**: We conduct additional experiments to investigate this question. Based finished experiments, we present some results below to respond to this question. We will add comprehensive results and analysis in our paper.
>
> **Multi-goal Maze:**
>
> We run DrAC with different values of $\beta$ in the our multi-goal maze environments, the final performances in the Hard maze are presented as follows.
>
> **DrAmort**
>
> | beta | Success Rate | Reachable Modes | Robustness-Removel | Robustness-Obstacle |
> | --- | --- | --- | --- | --- |
> | 0.6 | 1.00 | 1.0 | 0.50 | 0.04 |
> | 0.7 | 1.00 | 1.4 | 0.56 | 0.01 |
> | 0.8 | 1.00 | 4.6 | 0.91 | 0.62 |
> | 0.9 | 0.41 | 7.4 | 0.60 | 0.24 |
> | 1.0 | 0.36  | 7.4 | 0.55 | 0.23 |
>
> **DrDiffus**
>
> | $\beta$ | Success Rate | Reachable Modes | Robustness-Removel | Robustness-Obstacle |
> | --- | --- | --- | --- | --- |
> | 0.6 | 1.00 | 1.6 | 0.58 | 0.02 |
> | 0.7 | 1.00 | 2.0 | 0.55 | 0.03 |
> | 0.8 | 1.00 | 2.4 | 0.65 | 0.21 |
> | 0.9 | 0.90 | 6.8 | 0.83 | 0.65 |
> | 1.0 | 0.45 | 7.2 | 0.57 | 0.25 |
>
> According to the results, success rate is not sensitive to $\beta$ within a wide range  (about <0.8 for DrAmort and <0.9 for DrDiffus). diversity-related performance is sensitive to $\beta$. Higher $\beta$ values enable DrAC to reach more modes and enhance few-shot robustness. However, overly high temperatures reduce the success rate, which also harms robustness metrics. We will discuss the sensitivity of beta comprehensively in our revised version.
>
> **Game Content Generation:**
>
> We have run DrAC with three temperatures. The results for MarioPuzzle are displayed as follows
>
> **DrAmort**
>
> | $\beta$ | Content Quality | Content Diversity |
> | --- | --- | --- |
> | 0.2 | 30.9 | 631.3 |
> | 0.5 | 29.9 | 1137.3 |
> | 0.8 | 32.8 | 1384.9 |
>
> **DrDiffus**
>
> | $\beta$ | Content Quality | Content Diversity |
> | --- | --- | --- |
> | 0.2 | 35.3 | 744.3 |
> | 0.5 | 23.1 | 1061.3 |
> | 0.8 | 21.1 | 1009.3 |
>
> Regarding DrAmort, the diversity of generated levels increase along with beta raises. The content quality appears not sensitive to $\beta$.
>
> Regarding DrDiffus, when raising $\beta$ from 0.2 to 0.5, quality decreases and  diversity increase. When raising $\beta$ from  0.5 to 0.8, the performance does not change much.
>
> There are two aspects may be related to this experiment phenomenon: (1) higher temperature also improves exploration which could be helpful to quality (2) policy diversity is correlated but not identical to the diversity of generated levels. More fine-grained ablation study will be conducted and discussed in our revision. Future work in generative RL for game content generation includes multi-objecive RL, where our method could serve as a promising base algorithm.
>
> **MuJoCo**
>
> We are running additional experiments to analyze the sensitivity of DrAC in MuJoCo. Such experiments will be added in our revised version to improve thoroughness. We have finished partial experiments now. The performances of DrAmort with different values of $\beta$, in the first two MuJoCo environments, are presented as follows
>
> | $\beta$ | Ant | HalfCheetah |
> | --- | --- | --- |
> | 0.15 | 2513.8 | 5487.0 |
> | 0.25 | 3327.9 | 5640.9 |
> | 0.35 | 3440.2 | 6558.2 |
> | SAC | 3422.7 | 5515.7 |
>
> We also include the performance of SAC for reference. According to the table, raising $\beta$ value (as we used $\beta=0.15$ in the paper) improves the performance in Ant and HalfCheetah. Notably, the improvement in Ant from $\beta=0.15$ to $\beta=0.35$ is >36%, and with $\beta=0.35$, DrAmort outperforms SAC by >18% in HalfCheeta. More elaborate self-adaptation/annealing techniques could be researched in future work.
>
> **Q4**: The blue one ($X$) is a diagnoal Gaussian distribution $\mathcal{N}(\boldsymbol{0}, I)$, while the red one is a Gaussian mixture model with uniform weights and two components  $\{\mathcal{N}([6, 6], 0.2I),\mathcal{N}([-6, -6], 0.2I)\}$. We have modified the figure to describe the two ground truth distributions in our paper.
>
> **Q5**: The actor in SAC outputs the mean and per-dimension scale of a diagonal Gaussian distribution and applies $\tanh$ to bound the output action. It is closed to the concept of amortized inference. In the paper of SQL [1] and S2AC [2], the term “amortized actor” is used to refer to GAN-like actors, which take an additional random variable as input and output the action. So we follows there naming, specifically refer to SQL-style actor as “amortized actor”, which may be narrower than the concept of amortized inference. For SAC’s actor, we refer to it as “Gaussian actor”.
>
> We define “stochastic-mapping actors” as the framework to unify amortized actors and diffusion actors. This framework is also compatible with SAC-like actor, by treating $f_\theta(s,z) = \tanh(\mu_\theta(s) + \sigma_\theta(s)z), z \sim \mathcal{N}(\boldsymbol{0}, I)$, where $\mu_\theta(s)$ and $\sigma_\theta(s)$ is the mean vector and scale vector outputs by the neural network parameterized with $\theta$. That also means it will be straight-forward to apply our algorithm to train Gaussian actors. We did not test this as our main focus is training intractable multimodal actors. We will discussed this compatibility with Gaussian actor in our revision.
>
> [1] Haarnoja, Tuomas, et al. "Reinforcement learning with deep energy-based policies." *International conference on machine learning*. PMLR, 2017.
>
> [2] Messaoud, Safa, et al. "S$^2$AC: Energy-Based Reinforcement Learning with Stein Soft Actor Critic." *The Twelfth International Conference on Learning Representations*.
>
> **Q6**:  Thank for the careful review. We have updated Section 3.2 to acknowledge foundational works properly.
>
> **Regarding specific weaknesses:**
>
> >In the MuJoCo experiments ... It's unclear whether this is due to insufficient hyperparameter tuning or if multimodal policies simply aren't necessary in these environment
>
> We are addressing this by additional experiments (See response to question 2  and 3).
>
> >The process of tuning the temperature hyperparameters in the multi-goal experiment is not elaborated, potentially hindering the validity and rigorousness of the results.
>
> We address this in the response to question 1.
>
> >One of the claimed contributions, ... to accurately establish its place within the field.
>
> We thank the reviewer for pointing out this related work, as we were not aware of it before. (Lan et al 2022) has discussed the reparameterized policy gradient as an additional discovery, where the conclusion equation is equivalent to our Equation (7) in Proposition 1. We have revised Section 4.1 to cite Lan et al 2022 and re-clarify the contributions. We also revised our contribution statement 1 in the introduction.
>
> After reviewing related works more comprehensively, we would like to re-clarify our contribution as follows
>
> 1. Amortized actors are lightweight actors with strong multimodal expressiveness. However, they are underexplored in the field of multimodal RL. Previous practices in training amortized actors are based on SVGD [1,2]. The applicability of the reparameterized policy gradient for training amortized actors has not been formally discussed. We highlight that the reparameterized policy gradient is effective for training amortized actors.
> 2. Though there have been some diffusion policy-based RL algorithms back-propagating the gradient of the Q-function [3,4] to update the diffusion policy, the theoretical perspective from reparameterized policy gradient has not been formally discussed.
> 3. We formulate diffusion actors and amortized actors into a unified framework to show the versatility of reparameterized policy gradient for training intractable multimodal actors. This formulation serve as a useful tool for understanding and devising multimodal RL models and algorithms.
>
> [3] Wang, Yinuo, et al. "Diffusion actor-critic with entropy regulator." *Advances in Neural Information Processing Systems* 37 (2024): 54183-54204.
>
> [4] Ding, Zihan, and Chi Jin. "Consistency Models as a Rich and Efficient Policy Class for Reinforcement Learning." *The Twelfth International Conference on Learning Representations*.

---

> > ### Comment · Reviewer_T7pd · 2025-08-02
> > **Follow-up Questions**
> >
> > Thank you for the rebuttal, especially the additional analysis on the sensitivity to the temperature parameter. Quick clarification questions on your response to Q1: Other than the temperature parameter, are there other hyperparameters tuned in the grid search for each method? What ranges are used in the search?
> >
> > Also, to understand the empirical results a bit better, how is the learning rate for $alpha$ auto-adjustment selected? Is it based on some subset of environments?

---

> > > ### Author Response · Authors · 2025-08-03
> > >
> > > Thank you for your follow-up. Regarding these questions:
> > >
> > > 1. Only temperature parameters ($\beta$ for DrAC, target entropy for SAC and DACER, entropy coefficient for SQL and S2AC) are tuned during the grid search. Hyperparameters of other baselines follow the recommendation in their original papers/repositories.
> > > 2. The ranges of searching temperatures are determined based on our preliminary manual tuning experience. For both SAC and DACER, we used a range of $[0, 0.6|\mathcal{A}|]$ for searching their target entropy; for SQL, we used a range of $[0.1, 0.4]$ for searching its entropy coefficient; for S$^2$AC, we used a range of $[0.2, 0.8]$ for searching its entropy coefficient; for DrAC, we used a range of $[0.6, 1.0]$ for searching $\beta$.
> > > 3. Regarding the learning rate of $\alpha$: For SAC and DACER, we just follow the recommendation in their original paper for all environments. For our DrAC, we observed that when using 3e-4 (default in SAC), the actual (estimated) diversity fluctuates around the target diversity with relatively big amplitudes in all environments, so we roughly tuned up the learning rate of $\alpha$ to 5e-3 to reduce the fluctuations. We empirically found that the performance is not sensitive to the learning rate of $\alpha$, though we were unable to conduct a full set of experiments with multiple seeds regarding this. Generally, we would like to recommend 5e-3 as a default choice for DrAC.
> > >
> > > We will discuss hyperparameter choices and recommendations more comprehensively in our revised version.

---

> > > > ### Comment · Reviewer_T7pd · 2025-08-04
> > > >
> > > > Thank you for the response. Regarding the second point, since a grid search is used to tune the temperature, could you provide a sensitivity analysis, perhaps as a table at this phase, showing performance across different temperature values for each method?

---

> > > > > ### Author Response · Authors · 2025-08-06
> > > > >
> > > > > Thank you for your follow-up. We will add full sensitivity analysis in revision. We explain our temperature choice for baselines though a set of sensitivity analysis as follows.
> > > > >
> > > > > For **SAC**, in simple maze, we observed that the success rate reduces when raising target entropy from $0.5|\mathcal{A}|$ to $0.6|\mathcal{A}|$, so we chose a target entropy at $0.5|\mathcal{A}|$, i.e., $1.0$.
> > > > >
> > > > > | Target Entropy | Success Rate | Reachable Modes | Robustness-Removal | Robustness-Obstacle | Path Length |
> > > > > | ----------- | ------------ | --------------- | ------------------ | ------------------- | ----------- |
> > > > > | 0.8         | 100.0%       | 1.00            | 50.0%              | 12.7%               | 49.1        |
> > > > > | 1.0         | 100.0%       | 1.40            | 55.6%              | 16.6%               | 54.7        |
> > > > > | 1.2         | 75.0%        | 2.00            | 62.5%              | 47.0%               | 93.2        |
> > > > >
> > > > > For **DACER**, in hard maze, we observed that the success rate reduces when raising target from $0.5|\mathcal{A}|$ to $0.6|\mathcal{A}|$, so we chose target entropy at $0.5|\mathcal{A}|$. We observed that the estimated entropy does not reach the target when using a target entropy at $0.6|\mathcal{A}|$. Its noise scale increases unlimitedly, making the path length significantly increase and success rate decrease.
> > > > >
> > > > > | Target Entropy | Success Rate | Reachable Modes | Robustness-Removal | Robustness-Obstacle | Path length |
> > > > > | ----------- | ------------ | --------------- | ------------------ | ------------------- | ----------- |
> > > > > | 0.8         | 100.0%       | 1.00            | 50.2%              | 3.2%                | 90.4        |
> > > > > | 1.0         | 99.8%        | 1.20            | 50.6%              | 5.2%                | 86.5        |
> > > > > | 1.2         | 84.0%        | 6.60            | 85.7%              | 50.0%               | 329.3       |
> > > > >
> > > > > For **S$^2$AC**, in simple maze, we observed that the success rate reduces when raising entropy coefficient from 0.6 to 0.8, so we chose 0.6.
> > > > >
> > > > > | Entropy Coefficient | Success Rate | Reachable Modes | Robustness-Removal | Robustness-Obstacle | Path Length |
> > > > > | ----------- | ------------ | --------------- | ------------------ | ------------------- | ----------- |
> > > > > | 0.4         | 100.0%       | 1.67            | 70.2%              | 16.3%               | 54.8        |
> > > > > | 0.6         | 100.0%       | 2.60            | 96.0%              | 22.0%               | 67.7        |
> > > > > | 0.8         | 73.3%        | 2.67            | 90.8%              | 88.3%               | 121.7       |
> > > > >
> > > > > For **SQL**, in simple maze, the success rate start to decrease from an entropy coefficient at 0.4. As 0.3 and 0.4 do not show advantages against 0.2, we adopted 0.2.
> > > > >
> > > > > | Entropy Coefficient | Success Rate | Reachable Modes | Robustness-Removal | Robustness-Obstacle | Path Length |
> > > > > | ----------- | ------------ | --------------- | ------------------ | ------------------- | ----------- |
> > > > > | 0.2         | 100.0%       | 4.00            | 97.0%              | 65.8%               | 71.6        |
> > > > > | 0.3         | 100.0%       | 4.00            | 99.2%              | 57.4%               | 71.2        |
> > > > > | 0.4         | 99.5%        | 3.60            | 96.3%              | 63.4%               | 74.8        |
> > > > > | 0.45        | 98.0%        | 4.00            | 96.9%              | 71.2%               | 79.9        |
> > > > > | 0.6         | 79.0%        | 3.60            | 89.4%              | 78.4%               | 99.3        |

---

> > > > > > ### Comment · Reviewer_T7pd · 2025-08-09
> > > > > >
> > > > > > Thank you for providing the results for different temperatures. My main concerns around hyperparameter tuning and sensitivity to temperature are resolved. I believe with the new results and analysis promised in the rebuttal, the paper has become stronger. As a result, I am adjusting the rating to 5.

---

> > > > > > > ### Author Response · Authors · 2025-08-09
> > > > > > >
> > > > > > > Thank you for your constructive suggestions again! We will add full results to address the points raised during rebuttal. We believe this will significantly improve the rigorousness and clarity of our paper.

---

### Note · Authors · 2025-08-13

We thank all reviewers for their careful reviews and constructive comments. We believe that we have resolved most concerns with additional results. Key points are summarized below.

Reviewers recognized that we "proposes a novel loss to encourage diversity for multimodal policies and demonstrates its effectiveness" (Reviewer T7pd); "contains diverse experiments, demonstrating that DrAC elicits multimodal behaviours" (Reviewer yEqZ); our method "works well on a range of tasks", "often outperforms strong baselines" (Reviewer C7Lq); and our benchmarks are "more proper to evaluate the multimodal policy" (Reviewer 76hu).

Several common concerns were raised, and we have addressed them during the rebuttal and discussion:

**Hyperparameter Sensitivity Analysis:**

Most reviewers considered comprehensive ablation studies necessary to enhance experimental rigor. We have presented additional results proving that our comparisons were conducted with fair temperature hyperparameter configurations. We also demonstrated that our method produces better non-dominated policy populations in the game content generation environment. Most reviewers found these new results resolved their concerns. We will include full ablation studies in our revision.

**Statement of Proposition 1:**

Reviewers noted that some related theorems of our Proposition 1 were not well discussed, and its novelty was unclear. Since our main purpose is to highlight reparameterized policy gradient as a general tool for learning intractable multimodal actors, we will rewrite Proposition 1 as a single equation, discuss previously missed references, and rephrase related sentences in Section 4.2.

**Performance of amortized actor and diffusion actor:**

Several reviewers were surprised that the amortized actor outperformed the diffusion actor. While we understand this finding is counter-intuitive, to our knowledge, there is no evidence suggesting that amortized actors should perform worse than diffusion actors in online RL. Since we have ensured fair comparisons, we would like to pose this as an open question for future research.

**Other concerns:**

We have also addressed other specific concerns during the rebuttal, including comparisons with those baselines using tractable multimodal actors or without diversity bonuses, additional computational costs, and the notion of diversity. For missed references mentioned by reviewers, we will discuss them in our revision.

---

### Decision · Program_Chairs · 2025-09-17

**Decision:**

Accept (poster)

**Comment:**

This paper introduces Diversity-Regularized Actor-Critic (DrAC), a method for learning multimodal policies in continuous control settings. DrAC enforces diversity using a distance-based regularization loss and is implemented with two types of actors, i.e., amortized (DrAmort) and diffusion-based (DrDiffus). While DrAC's performance is on par with SAC on standard MuJoCo benchmarks, it demonstrates clear benefits in multi-goal and generative tasks where policy diversity is essential.

Reviewers found the paper to be a valuable exploration of expressive policies in understudied but important domains. During the rebuttal, the authors provided additional experiments that successfully addressed most initial concerns regarding hyperparameter sensitivity and the comparison between the DrAmort and DrDiffus variants.

A concern shared across multiple reviewers remains regarding the observation that the amortized actor outperforms the diffusion actor. While the authors correctly state there is no prior evidence to suggest the contrary, the paper would be strengthened by providing more intuition or analysis to explain this performance difference. Nevertheless, the overall contribution is solid and the paper is recommended for acceptance.